# ScatterAD: Temporal-Topological Scattering Mechanism for Time Series Anomaly Detection

**Tao Yin**[1]    **Xiaohong Zhang**[1]*    **Shaochen Fu**[1]    **Zhibin Zhang**[1]    **Li Huang**[1]
**Yiyuan Yang**[2]    **Kaixiang Yang**[3]    **Meng Yan**[1]*

[1]Chongqing University    [2]University of Oxford    [3]South China University of Technology

{yintao, fushaochen}@stu.cqu.edu.cn, {xhongz, zbinz, lee.h, mengy}@cqu.edu.cn
yiyuan.yang@cs.ox.ac.uk, yangkx@scut.edu.cn

## Abstract

One main challenge in time series anomaly detection for industrial IoT lies in the complex spatio-temporal couplings within multivariate data. However, traditional anomaly detection methods focus on modeling spatial or temporal dependencies independently, resulting in suboptimal representation learning and limited sensitivity to anomalous dispersion in high-dimensional spaces. In this work, we conduct an empirical analysis showing that both normal and anomalous samples tend to scatter in high-dimensional space, especially anomalous samples are markedly more dispersed. We formalize this dispersion phenomenon as scattering, quantified by the mean pairwise distance among sample representations, and leverage it as an inductive signal to enhance spatio-temporal anomaly detection. Technically, we propose ScatterAD to model representation scattering across temporal and topological dimensions. ScatterAD incorporates a topological encoder for capturing graph-structured scattering and a temporal encoder for constraining over-scattering through mean squared error minimization between neighboring time steps. We introduce a contrastive fusion mechanism to ensure the complementarity of the learned temporal and topological representations. Additionally, we theoretically show that maximizing the conditional mutual information between temporal and topological views improves cross-view consistency and enhances more discriminative representations. Extensive experiments on multiple public benchmarks show that ScatterAD achieves state-of-the-art performance on multivariate time series anomaly detection. Code is available at this repository: https://github.com/jk-sounds/ScatterAD.

## 1   Introduction

Multivariate time series data in Industrial Internet of Things (IoT) systems exhibit intricate spatio-temporal relationships[Tian et al., 2023], and anomaly detection is crucial to ensure the stable operation of the system[Efthymiou et al., 2018, Baraniuk, 2020]. Recognizing abnormal patterns from complex data has become a critical and widely studied problem in both research and practical domains[Mohammadi et al., 2018, Wollschlaeger et al., 2017]. However, anomalies in multivariate time series frequently manifest synergistically in temporal and topological dimensions[Jin et al., 2024, Yi et al., 2024, Zhao et al., 2024], presenting highly coupled complex characteristics, which pose a severe challenge to traditional anomaly detection methods.

---

*Corresponding authors.

39th Conference on Neural Information Processing Systems (NeurIPS 2025).

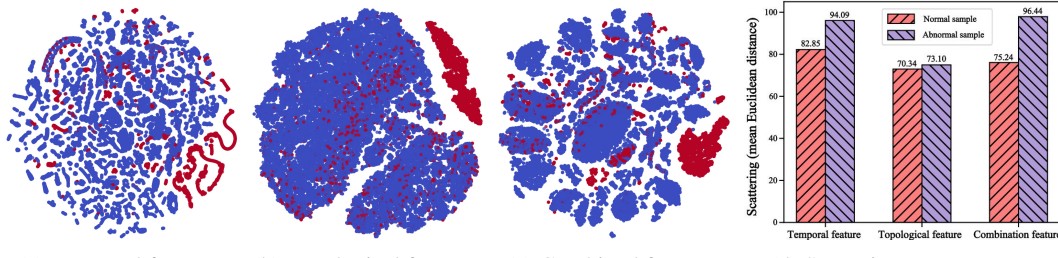

| (a) Temporal features | (b) Topological features | (c) Combined features | (d) Scattering measurement |

Figure 1: The distribution patterns of scatter visualization and scatter measurement in time features, topological features, and combined temporal-topological features (obtained by simple linear combination) on the SWaT dataset. Blue and red represent the embeddings of normal points and abnormal points respectively. Normal data exhibits clustering characteristics, abnormal data exhibits scattering characteristics, and combined features show more significant scattering differences. This provides a strong prior signal for anomaly detection.

Time series anomaly detection methods are broadly bifurcated into two principal categories: methods based on temporal pattern learning and methods based on topological structure modeling. Temporal pattern-based methods (such as[Park et al., 2018, Su et al., 2019, Xu et al., 2021, Wu et al., 2021]) primarily focus on the sequential dependency of time series data and identify data points that deviate from normal temporal patterns by constructing predictive or reconstructive models. However, recursive models for learning temporal patterns exhibit limitations in capturing pairwise dependencies between variables, limiting their ability to identify intricate anomalies[Zhao et al., 2020, Xu et al., 2021]. Conversely, topological structure-based methods (such as GDN[Deng and Hooi, 2021], GTAD[Guan et al., 2022], TopoGDN[Liu et al., 2024] etc.) focus on modeling the graph structure relationship between variables and detecting anomalies by learning the mutual dependence between variables. While these methodologies can model spatial correlations, they often fail to fully leverage the dynamic characteristics of time series. Methodologies focusing exclusively on either temporal or spatial patterns often prove inadequate for modeling the intricate spatio-temporal dynamics in multivariate time series, consequently leading to challenges in effectively capturing the "scattering" phenomenon of anomalies in the spatio-temporal dimension. As shown in Figure 1, we visualize the scattering patterns of temporal features, topological features, and their linear combination features on the SWaT dataset, using the average Euclidean distance to quantify the degree of scattering between samples. It is evident from the scattering representation diagram and the scattering metric diagram that the combined feature space usually exhibits more pronounced scattering differences than the temporal feature space and the topological feature space. In essence, anomaly detection is the identification of samples outside the data distribution (Out-of-distribution, OOD)[Wang et al., 2020]. Significant scattering differences can bring better anomaly separation capabilities to the model. In other words, when the scattering of anomalous samples is more pronounced, their feature representations are pushed further apart from the normal samples, creating a clearer boundary for anomaly detection. More recently, several studies have endeavored to consider both the temporal and spatial dimensions, such as MTAD-GAT[Zhao et al., 2020]. However, MTAD-GAT employ simple feature concatenation or serial processing methods and fail to address the challenge of collaborative optimization of spatio-temporal features in theory. In particular, they lack a deep understanding and modeling of the complementarity of spatio-temporal features.

The information bottleneck principle[Tishby et al., 2000] offers a theoretical framework for addressing such challenges of spatio-temporal feature complementarity. This principle aims to extract a representation Z from the input variable X, such that Z preserves maximal information as possible about the target variable Y in X, while compressing redundant information in X that is irrelevant to Y. In particular, when complementary representations need to be learned from different dimensions (such as temporal and topological), a single bottleneck objective may fail to effectively align the relationship between the representations of each dimension[Federici et al., 2020]. Considering that anomalies in multivariate time series typically manifest as abnormal scattering of spatio-temporal patterns, we propose a theoretical hypothesis: effective anomaly detection needs to capture the scattering representation characteristics of both the temporal and topological dimensions and ensure the complementarity of these two features. Through theoretical analysis, we prove that under the condition of a given graph

structure G, maximizing the mutual information $I(Z_T; Z_G \mid G)$ between the temporal representation $Z_T$ and the topological representation $Z_G$ can facilitate effective anomaly detection. This theoretical result can be expressed as: $\max I(Z_T; Z_G \mid G)$ s.t. $I(X, Z_T) < r_1, I(X, Z_G) < r_2$, where $r_1$ and $r_2$ control the compression degree of the temporal and topological representations respectively. This shows that, subject to the constraint that the temporal and topological representations are fully compressed, maximizing their conditional mutual information can promote the synergistic complementarity of the two representations, thereby improving the performance of anomaly detection.

Specifically, we propose a novel anomaly detection approach ScatterAD, which enables effective representation learning of spatio-temporal representation through three principal mechanisms: first, the scattering mechanism is designed to capture the scattering characteristics of the representation in the feature dimension in the temporal-topological representation; second, we further incorporate a time constraint mechanism to prevent excessive scattering of the representation to maintain the consistency of the temporal structure; finally, in order to align temporal and topological representations in the latent representation space, we introduce contrastive fuse mechanism to align the outputs of the temporal and topological encoders, maintaining the compression constraints while promoting the complementarity of the representations. Unlike prior work, which treats temporal and topological patterns separately, ScatterAD introduces a unified contrastive scattering mechanism that jointly optimizes temporal consistency and structural distinguishability.

The contribution of our paper is summarized as follows:

- We introduce ScatterAD, a novel anomaly detection approach that employs a temporal-topological scattering mechanism to improve representational discriminability while preserving temporal structural consistency.

- To the best of our knowledge, this is the first work to introduce information bottleneck theory into multivariate time series anomaly detection, to theoretically reveal the complementarity between temporal and topological features. This leads to more discriminative representations and highlights the importance of integrating spatio-temporal information in future anomaly detection research.

- Extensive experiments conducted on six public benchmark datasets and using twelve standard evaluation metrics demonstrate that ScatterAD achieves state-of-the-art performance, validating both the theoretical foundations and practical effectiveness of our design.

## 2 Related Work

**Anomaly Detection in Time Series** Given the scarcity and imbalance of data labels, unsupervised anomaly detection methods have received widespread attention in recent years [Choi et al., 2021]. Primarily focused on the reconstruction error-driven autoencoder model [Kingma et al., 2013], or its probabilistic extension, the variational autoencoder [Su et al., 2019]. These methods capture local reconstruction patterns but struggle to capture fine-scale dependencies and dynamic propagation across time points. Graph neural networks (GNNs) have been introduced to model the dependency structure between different variables in multivariate time series. Representative methods such as MTGNN [Wu et al., 2020] and GDN [Deng and Hooi, 2021] typically regard each variable as a node in the graph and introduce graph convolution to learn spatial structure. However, these methods focus on spatial graph modeling between variables, neglecting the temporal graph structure. Consequently, they depend on topological associations between variables and have limited capacity to model the time dimension. More recently, several studies have initiated the exploration of graph structures in the time dimension. For instance, GANF [Dai and Chen, 2022] and MTGFlow [Zhou et al., 2023] construct time series graph structures and combine density modeling methods to model the distribution of the entire sequence, but these methods still do not deeply explore the structural characteristics and interdependencies between time nodes. Although methods such as TopoGDN [Liu et al., 2024] and CST-GL [Zheng et al., 2023] model spatio-temporal dependencies, they overlook representation scattering in high-dimensional spaces, which can potentially impede their sensitivity to the dispersion patterns characteristic of anomalies.

**Information bottleneck and representation complementarity** The classic information bottleneck principle [Tishby et al., 2000] explicitly articulates the objective of retaining task-relevant information while compressing redundant inputs. Based on this foundation, Deep InfoMax [Hjelm et al., 2018]

and InfoGraph [Sun et al., 2019] employ a mutual information maximization strategy to align local and global representations in structured data, including images and graphs. Models such as MVGRL [Hassani and Khasahmadi, 2020] and structured mutual information models [Yang et al., 2025] promote complementarity between different representation views, which have the potential to substantially enhance the performance of downstream tasks.

# 3 Method

## 3.1 Overview

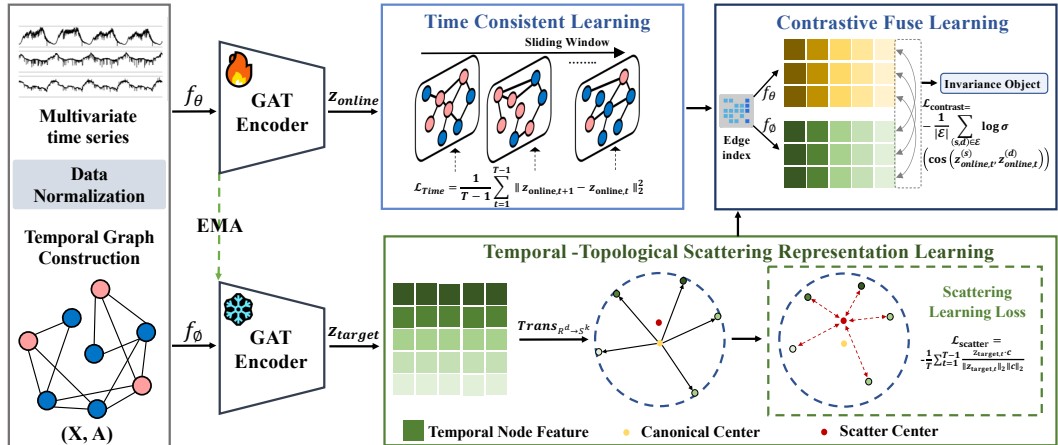

Figure 2: The overall framework of ScatterAD employs temporal graphs to model the scattering patterns of nodes in multivariate time series. It encodes temporal nodes as graph vertices to capture structural scattering, constrains time consistency to avoid excessive scattering, and aligns temporal topological features through contrastive fusion, ultimately achieving effective detection through scattering representation learning.

As illustrated in Figure 2, ScatterAD employs an online encoder $f_\theta(\cdot)$ and a target encoder $f_\phi(\cdot)$ to process the temporal graph $G$, where $\theta$ and $\phi$ denote the parameters of the respective encoders. (1) A topological space is initially defined via $L_2$-normalizing node features onto the unit hypersphere, with a global scatter center $c$ randomly initialized inside the unit ball (see Section 3.3 for details). Following the positive-only sampling strategy used in DCdetector [Yang et al., 2023], all training samples are treated as positive samples. Within this embedding space, the target encoder $f_\phi(\cdot)$ learns to model the compactness of normal samples, thereby capturing scattering representations and amplifying the scattering of anomalous samples during inference. (2) The online encoder $f_\theta(\cdot)$ yields $h_{\text{online}}$ and enforces temporal smoothness via a temporal constraint on adjacent temporal nodes, preventing excessive scattering in the target encoder $f_\phi(\cdot)$. Only the online encoder is updated via backpropagation; the target encoder parameters $\phi$ are updated using the exponential moving average (EMA) of the online encoder parameters $\theta$ after each training step (see Section 3.2 for details). (3) The online and target encoders constitute a contrastive fusion framework to maximize the similarity between them, thereby fostering collaboration between the online and target encoders. (4) Regarding the anomaly judgment criterion, anomalies are evaluated based on scattering deviation and time consistency deviation (see Section 3.3 for details).

The definition of multivariate time series anomaly detection is as follows: Let $\mathcal{X} = \{x_t\}_{t=1}^T$, $\mathcal{X} \in \mathbb{R}^{T \times N}$ be a multivariate time series with N dimensions and a sliding window length of T, $x_t = [x_t^1, x_t^2, \ldots, x_t^N]^\top \in \mathbb{R}^N$ represent the N-dimensional observation vector at time t, and the sequence length is T. The goal of anomaly detection is to predict the anomaly label $\mathcal{Y}_t \in \{0, 1\}$ for each timestamp t, where $\mathcal{Y}_t = 1$ indicates that time t is in an abnormal state, and $\mathcal{Y}_t = 0$ indicates a normal state.

## 3.2 Temporal Graph Construction and Node Feature Extraction

Our framework employs a dual-encoder architecture to learn temporally and topologically aggregated representations of normal patterns at temporal nodes. Unlike prior graph-based time series models, where each node corresponds to a sensor or univariate stream, we consider each time step as a distinct node. Specifically, each graph node represents a multidimensional time slice $x_t \in \mathbb{R}^N$, and the edge set is defined as $\mathcal{E} = (v_{t-k}, v_t) \mid k \in [1, \tau], t \in [\tau + 1, T]$, capturing the behavioral patterns across time through directed connections. Here, $\tau$ is the look-back window size. In our experiments, we set $\tau = 2$. To extract temporal dynamics from the input sequence, we apply a multi-scale causal convolutional encoder. Let $h^{(0)} = \mathcal{X}$ denote the input time series:

$$h^{(l)} = \text{PReLU}\left(\mathcal{BN}\left(\sum_{i=0}^{k-1} \mathcal{W}_c^{(i)} \cdot h_{t-i}^{(l-1)}\right)\right), \quad \text{for } t = k, \ldots, \mathcal{T}, \tag{1}$$

where $\mathcal{W}c^{(i)}$ denotes the $i$-th convolution kernel in a $k$-length causal convolution, $\mathcal{BN}(\cdot)$ signifies batch normalization [Ioffe and Szegedy, 2015]. To further integrate topological dependencies across temporal nodes, we employ a multi-head GAT layer to compute attention coefficients across nodes:

$$\alpha_{ij}^h = \frac{\exp\left(\text{LeakyReLU}(a_h^\top [\mathcal{W}_h h_i \| \mathcal{W}_h h_j])\right)}{\sum_{k \in \mathcal{N}_i} \exp\left(\text{LeakyReLU}(a_h^\top [\mathcal{W}_h h_i \| \mathcal{W}_h h_k])\right)}, \tag{2}$$

$\alpha_{ij}^h$ denotes the normalized attention coefficients of the h-th head. $h_i, h_j \in h^{(l)}$ represent the input feature vectors of nodes i and j. $\mathcal{W}_h$ is the head-specific parameter matrix and the $a_h^\top$ is the attention coefficient vector.

$$h_i' = \text{ELU}\left(\bigoplus_{h=1}^{\mathcal{H}} \left(\sum_{j \in \mathcal{N}_i} \text{softmax}\left(a_{ij}^{(h)}\right) \mathcal{W}_h h_j\right)\right). \tag{3}$$

Subsequently, the node features are aggregated with topological features and temporal features, yielding $z_i = h_i' + h^{(l)}$. $z_i \in \mathbb{R}^{T \times d_{out}}$ denotes the stacked temporal and topological representations.

## 3.3 Optimization Target

**Time Consistent Learning**   The loss of time consistency is introduced to node features $z_{online,t} \in z_i$ to reinforce the temporal dependency of adjacent nodes and prevent them from excessive dispersion in the feature subspace, thereby preserving the key temporal features of the nodes. The formulation is as follows:

$$\mathcal{L}_{time} = \frac{1}{\mathcal{T}-1} \sum_{t=1}^{\mathcal{T}-1} \|z_{online,t+1} - z_{online,t}\|_2^2, \tag{4}$$

here, each row $z_{online,t} \in \mathbb{R}^{d_{out}}$ corresponds to the node representation at time step $t$ and $\mathcal{L}_{time}$ penalizes abrupt representation shifts between consecutive encoding states.

**Temporal Topological Scattering Representation Learning**   We propose a graph scattering mechanism to explicitly guide the target encoder in learning decentralized representations of nodes. For node representation, we initially define a regular subspace $\mathbb{S}^k$ and a scattering center $c$. For the subspace $\mathbb{S}^k$, a transformation function $Trans(\cdot)$ is employed to transform the representation from the original space $\mathbb{R}^d$ to $\mathbb{S}^k$. Specifically, we apply $L_2$-normalizing to each row vector $z_i$ within the matrix $z_{target,t}$:

$$\widetilde{z}_{target,t} = Trans_{\mathbb{R}^d \to \mathbb{S}^k}(z_{target,t}) = \frac{z_{target,t}}{\|z_{target,t}\|_2}, \quad \mathbb{S}^k = \left\{\widetilde{z}_{target,t} : \|\widetilde{z}_{target,t}\|_2 = 1\right\}, \tag{5}$$

where $\widetilde{z}_{target,t} \in \mathbb{S}^k$ is the target representation on the unit sphere (normalized), as defined in the formula, the representations of all nodes are distributed in the hypersphere $\mathbb{S}^k$. This mapping prevents arbitrary scattering of representations in the space, thereby mitigating instability and optimization difficulties during training. Afterwards, we initialize a global scattering center, **c**, which serves as a fixed anchor point within the latent space. This center is randomly sampled to reside strictly

inside the unit hypersphere. The initialization is a two-step process: first, an intermediate vector $\mathbf{c}'$ is sampled from a standard multivariate normal distribution. Second, this vector is L2-normalized to project it onto the unit hypersphere and then scaled by a random scalar $\epsilon \in (0, 1)$. This procedure can be formally expressed as: $\mathbf{c}' \sim \mathcal{N}(0, I), \quad \mathbf{c} = \frac{\mathbf{c}'}{\|\mathbf{c}'\|_2} \cdot \epsilon, \quad$ where $\epsilon \sim \mathcal{U}(0, 1)$. With this fixed scattering center $\mathbf{c}$, we then introduce the representation scattering loss, $L_{\text{scatter}}$, designed to pull the representations of normal samples towards it:

$$\mathcal{L}_{scatter} = -\frac{1}{T} \sum_{t=1}^{\mathcal{T}} \frac{\widetilde{z}_{target,t}^{\top} \cdot c}{\|\widetilde{z}_{target,t}\|_2 \|c\|_2}. \tag{6}$$

By maximizing this loss function, the learned target features remain close to the semantic center while preserving directional diversity, which is crucial for amplifying the scattering behavior of anomalous nodes during inference.

**Contrastive Fuse Learning**    To ensure that the nodes maintain consistent directions in both the embedding space and the projected hypersphere space, while preserving local structures and global geometric properties. We maximize the similarity between positive sample pairs from both encoders, thereby ensuring that the embedding space preserves local structural relationships while the scattering space maintains global node relationships. The formulation is as follows:

$$\mathcal{L}_{contrast} = -\frac{1}{|\mathcal{E}|} \sum_{(s,d) \in \mathcal{E}} \log \sigma \left( cos \left( z_{online,t}^{(s)} \cdot z_{target,t}^{(d)} \right) \right). \tag{7}$$

Here, $\mathcal{E}$ denotes the set of edges of graph and $\sigma$ is the sigmoid activation function, $(s, d)$ representing the origin and terminus of the corresponding edge, constituting a positive sample pair, $z_{online,t}^{(s)}$ and $z_{target,t}^{(d)}$ denote the feature vectors generated by the encoder $f_\theta(\cdot)$ and the encoder $f_\phi(\cdot)$.

**Exponential Moving Average Mechanism**    During the training process, the parameters $\theta$ of the online encoder are updated via gradient descent, and the parameters $\phi$ are updated through the Exponential Moving Average mechanism [Cai et al., 2021] by the online encoder. Compared with direct alignment constraints and centralized representations, the prediction target serves as a buffer mechanism, enabling the target encoder to adaptively learn local time constraint information and global topological scattering representation. The exponential moving average mechanism is applied at the end of each training epoch. The formula is as follows:

$$\theta \leftarrow m\theta + (1 - m)\xi. \tag{8}$$

Among them, $(0 < m < 1)$, this process facilitates the integration of graph time structure consistency into the node scattering representation process, effectively alleviating the confrontation brought about by direct optimization of alignment constraints and decentralized representations. Representation scattering is used to enhance the global distinguishability of information, and time consistency loss ensures the consistency of local information. The interaction of these two mechanisms enables the target encoder to learn global topological scattering features more stably while preserving local temporal information.

**Loss Function**    The overall training objective of ScatterAD integrates three loss components:

$$\mathcal{L}_{total} = \mathcal{L}_{scatter} + \mathcal{L}_{time} + \mathcal{L}_{contrast}, \tag{9}$$

$\mathcal{L}_{scatter}$ captures the scattering representation by modeling the compactness of normal samples within the topological space, thereby implicitly amplifying the contrast with anomalous patterns and enhancing feature discriminability. $\mathcal{L}_{time}$ enforces time consistency by penalizing abrupt changes between adjacent time steps, mitigating over-scattering. $\mathcal{L}_{contrast}$ aligns temporal and topological representations through a cross-view contrastive loss based on cosine similarity, fostering their complementarity. This joint optimization encourages the model to learn discriminative, temporally consistent, and cross-view complementary representations for robust anomaly detection.

## 3.4    Anomaly Criterion

To jointly evaluate anomalies from both the topological scattering and time consistency perspectives, we define a unified anomaly scoring function that captures two complementary forms of deviation.

For the anomaly score of the input sequence $\mathcal{X}$, it is initially transformed into a node representation $z_i = f_\theta(\mathcal{X})$, $z_i \in \mathbb{R}^{N \times d_{out}}$. The scattering deviation score $\mathcal{D}(\mathcal{X})$, which quantifies how far the target node representation deviates from the learned center in the normalized representation space, and the time inconsistency score $\mathcal{L}(\mathcal{X})$, which penalizes abrupt representation shifts between consecutive time steps. The ultimate anomaly score is formulated as follows:

$$\text{AnomalyScore}(\mathcal{X}) = \underbrace{\mathcal{D}(\mathcal{X}) : \frac{1}{\min_k \mathcal{D}(z_i, c)}}_{Scattering \quad deviation} + \underbrace{\mathcal{L}(\mathcal{X}) : \frac{1}{d} \sum_{i=1}^{d} \left( z_i^{(t)} - z_i^{(t-1)} \right)^2}_{Time \quad inconsistency}. \quad (10)$$

This is a point-level anomaly detection approach, where anomalous points are expected to receive higher anomaly scores. Following prior work [Xu et al., 2021, Yang et al., 2023], we introduce a threshold $\delta$ to convert scores into binary labels: a point is labeled as anomalous ($y = 1$) if its score exceeds $\delta$, and normal ($y = 0$) otherwise.

## 4 Experiments

### 4.1 Experimental Setup

**Datasets**

We conducted evaluations on six real-world multivariate time series datasets, including **(1) PSM** (Pooled Server Metrics)[Abdulaal et al., 2021]; **(2) MSL** (Mars Science Laboratory)[Hundman et al., 2018]; **(3) SWaT**(Secure Water Treatment)[Mathur and Tippenhauer, 2016]; **(4) WADI** (Water Distribution)[Ahmed et al., 2017]; **(5) NIPS-TS-SWAN**[Lai et al., 2021]; **(6) NIPS-TS-GECCO**[Lai et al., 2021]. See Appendix B for dataset details and statistics.

**Metrics**  We evaluate detection performance using both label-based and score-based metrics. For label-based metrics, we report the point-adjusted F1 score (PA-F), which considers an anomaly segment detected if any timestamp within it is flagged. Although it may overestimate performance, it is widely used [Xu et al., 2021, Guan et al., 2022, Song et al., 2023]. To ensure fair comparisons, we also report the Affiliated F1 score (Aff-F) [Huet et al., 2022], which measures alignment quality between predicted and ground-truth anomaly segments. Additionally, we include score-based metrics: Area under the ROC Curve (A-ROC) [Fawcett, 2006] and Area under the Precision-Recall Curve (A-PR)[Boyd et al., 2013]. In summation, we report results across twelve evaluation metrics, with comprehensive reports detailed reports in Appendix C.

**Baselines**  We compare our method against a diverse set of baselines, including recent state-of-the-art graph-based and transformer-based models: Sub-Adjacent Transformer (S.T.)[Yue et al., 2024], TopoGDN (T.G.)[Liu et al., 2024], Memto [Song et al., 2023], DuoGAT (D.G.)[Lee et al., 2023], MTGFlow (MTG)[Zhou et al., 2023], iTransformer (iT.)[Liu et al., 2023], DCdetector (DC.)[Yang et al., 2023], Anomaly Transformer (A.T.)[Xu et al., 2021], and GANF[Dai and Chen, 2022]. We also include classical methods such as ModernTCN (M.TCN)[Luo and Wang, 2024], Variational Autoencoder (VAE)[Pinheiro Cinelli et al., 2021], Isolation Forest (IF)[Liu et al., 2008], and PCA [Shyu et al., 2003] for a comprehensive evaluation.

### 4.2 Main Results

We evaluate ScatterAD on six real-world multivariate time series datasets utilizing four standard evaluation metrics: Affiliated-F1 (Aff-F), Point-Adjusted-F1 (PA-F), AUC-ROC (A-ROC), and AUC-PR (A-PR). As demonstrated in Table 1, ScatterAD consistently achieves the best performance, ranking first in 15 out of 24 evaluation settings. Notably, ScatterAD achieves the highest Affiliated-F1 scores across all datasets, demonstrating its superior ability to localize anomalies at the segment level. While performance on the NIPS-TS-SWAN (N-T-S) dataset is mixed, with the highest scores in PA-F (0.736), A-ROC (0.792), and A-PR (0.716), but a notably low Aff-F (0.038), we attribute this to the irregular and bursty nature of anomalies in the dataset, which makes segment-level matching particularly challenging. Nonetheless, the robust Point-Adjusted-F1 suggests robust detection capabilities under noisy and complex conditions. Furthermore, ScatterAD consistently achieves the highest AUC-ROC values across all datasets, indicating a strong trade-off between true positive and false positive rates under varying threshold settings.

Table 1: Performance comparison on six real-world datasets. Evaluation Metrics: Aff-F (affiliated F1), PA-F (point-adjusted F1), A-ROC (AUC-ROC), and A-PR (AUC-PR). Higher values indicate better performance. All reported results are averaged over multiple independent runs to ensure robustness. **Bold** indicates the optimal performance; Underline indicates the suboptimal performance.

| Dataset | Metric | Ours | S.T. | T.G. | Memto | DC. | M.TCN | iT. | MTG | A.T. | D.G. | GANF | VAE | IF | PCA |
|---|---|---|---|---|---|---|---|---|---|---|---|---|---|---|---|
| **MSL** | Aff-F | **0.867** | 0.673 | 0.674 | 0.595 | 0.674 | 0.709 | 0.652 | 0.374 | 0.673 | 0.677 | 0.323 | 0.642 | 0.374 | 0.591 |
| | PA-F | **0.964** | 0.863 | 0.714 | 0.664 | 0.947 | 0.807 | 0.659 | 0.648 | 0.934 | 0.652 | 0.489 | 0.512 | 0.598 | 0.444 |
| | A-ROC | **0.986** | 0.751 | 0.794 | 0.951 | 0.961 | 0.627 | 0.604 | 0.857 | 0.970 | 0.788 | 0.678 | 0.883 | 0.673 | 0.732 |
| | A-PR | **0.932** | 0.754 | 0.727 | 0.902 | 0.891 | 0.739 | 0.721 | 0.781 | 0.878 | 0.703 | 0.620 | 0.521 | 0.519 | 0.524 |
| **PSM** | Aff-F | **0.797** | 0.786 | 0.689 | 0.659 | 0.653 | 0.701 | 0.652 | 0.436 | 0.657 | 0.624 | 0.329 | 0.524 | 0.569 | 0.437 |
| | PA-F | **0.981** | 0.941 | 0.801 | 0.977 | 0.977 | 0.965 | 0.926 | 0.797 | 0.978 | 0.780 | 0.788 | 0.596 | 0.543 | 0.467 |
| | A-ROC | **0.986** | 0.731 | 0.830 | 0.981 | 0.948 | 0.581 | 0.687 | 0.832 | 0.977 | 0.074 | 0.825 | 0.758 | 0.634 | 0.908 |
| | A-PR | **0.969** | 0.806 | 0.965 | 0.542 | 0.866 | 0.629 | 0.751 | 0.753 | 0.961 | 0.245 | 0.745 | 0.443 | 0.344 | 0.687 |
| **SWaT** | Aff-F | 0.704 | 0.631 | 0.594 | 0.592 | 0.687 | 0.685 | **0.716** | 0.463 | 0.622 | 0.619 | 0.608 | 0.563 | 0.502 | 0.537 |
| | PA-F | **0.951** | 0.863 | 0.788 | 0.756 | 0.939 | 0.884 | 0.916 | 0.500 | 0.943 | 0.822 | 0.244 | 0.526 | 0.472 | 0.548 |
| | A-ROC | 0.982 | 0.956 | 0.766 | 0.841 | 0.876 | 0.675 | 0.662 | 0.769 | **0.989** | 0.094 | 0.574 | 0.532 | 0.501 | 0.502 |
| | A-PR | **0.909** | 0.868 | 0.822 | 0.810 | 0.824 | 0.592 | 0.619 | 0.851 | 0.657 | 0.503 | 0.830 | 0.432 | 0.511 | 0.421 |
| **WADI** | Aff-F | 0.605 | **0.756** | 0.532 | 0.701 | 0.725 | 0.558 | 0.671 | 0.673 | 0.708 | 0.636 | 0.667 | 0.556 | 0.555 | 0.477 |
| | PA-F | 0.862 | **0.873** | 0.834 | 0.807 | 0.731 | 0.766 | 0.754 | 0.690 | 0.738 | 0.705 | 0.681 | 0.512 | 0.474 | 0.618 |
| | A-ROC | 0.960 | 0.945 | 0.612 | 0.693 | 0.829 | 0.831 | 0.814 | 0.782 | **0.982** | 0.372 | 0.782 | 0.501 | 0.860 | 0.502 |
| | A-PR | **0.765** | 0.743 | 0.532 | 0.691 | 0.651 | 0.697 | 0.627 | 0.523 | 0.637 | 0.470 | 0.508 | 0.384 | 0.406 | 0.357 |
| **NIPS-TS-SWAN** | Aff-F | 0.038 | **0.805** | 0.685 | 0.033 | 0.484 | 0.533 | 0.507 | 0.003 | 0.099 | 0.659 | 0.005 | 0.385 | 0.438 | 0.680 |
| | PA-F | 0.736 | 0.714 | 0.660 | 0.687 | 0.733 | 0.731 | 0.729 | 0.591 | 0.695 | 0.503 | 0.630 | 0.497 | 0.522 | 0.526 |
| | A-ROC | 0.792 | **0.870** | 0.595 | 0.791 | 0.655 | 0.562 | 0.537 | 0.788 | 0.787 | 0.671 | 0.778 | 0.522 | 0.729 | 0.498 |
| | A-PR | 0.716 | **0.946** | 0.527 | 0.702 | 0.626 | 0.545 | 0.531 | 0.699 | 0.569 | 0.714 | 0.712 | 0.326 | 0.473 | 0.326 |
| **NIPS-TS-GECCO** | Aff-F | **0.825** | 0.476 | 0.589 | 0.424 | 0.648 | 0.446 | 0.435 | 0.348 | 0.658 | 0.667 | 0.268 | 0.481 | 0.531 | 0.509 |
| | PA-F | **0.784** | 0.491 | 0.670 | 0.504 | 0.472 | 0.357 | 0.381 | 0.381 | 0.325 | 0.625 | 0.355 | 0.491 | 0.481 | 0.583 |
| | A-ROC | **0.969** | 0.536 | 0.787 | 0.875 | 0.715 | 0.946 | 0.817 | 0.732 | 0.685 | 0.690 | 0.898 | 0.868 | 0.684 | 0.763 |
| | A-PR | 0.633 | 0.318 | 0.495 | 0.519 | 0.523 | 0.615 | 0.474 | 0.552 | **0.870** | 0.535 | 0.337 | 0.443 | 0.418 | 0.543 |
| **#1 Count** | | 15 | 4 | 0 | 0 | 0 | 0 | 1 | 0 | 3 | 0 | 0 | 0 | 0 | 0 |

Figure 3 illustrates anomaly score distributions on SWaT. As shown in Figures 3a- 3c, competing methods exhibit significant overlap between normal and anomalous scores, limiting detection precision. In contrast, ScatterAD (Figure 3d) exhibits a clear separation, and the anomaly scores for normal points are sharply concentrated near zero, while anomalies are widely spread across higher scores, forming multiple distinguishable modes. This distinct gap between normal and anomalous distributions greatly reduces the false positive rate and enhances detection precision.

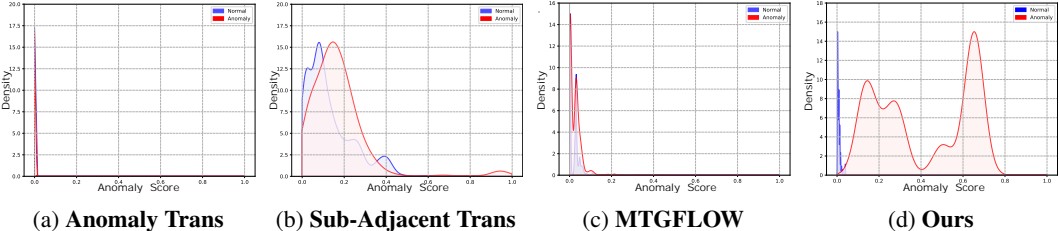

(a) **Anomaly Trans**    (b) **Sub-Adjacent Trans**    (c) **MTGFLOW**    (d) **Ours**

Figure 3: The distribution of anomaly scores presented by different frameworks on the SWaT dataset.

## 4.3 Ablation Studies

To validate the effectiveness of ScatterAD's components, we conduct an ablation study on its dual encoder architecture and contrastive fusion mechanism (Table 2). Removing the temporal online encoder ("w/o T-Enc ") impairs the model's ability to capture temporal variations, reducing detection performance. Excluding the topological target encoder ("w/o S-Enc ") leads to significant drops, particularly on WADI and SWAT, underscoring the need to model global structural dependencies. Eliminating the contrastive fusion module ("w/o C-Fuse") results in consistent performance degradation across all datasets, confirming its importance in aligning temporal and topological cues. Removing the Exponential Moving Average mechanism ("w/o EMA") destabilizes target encoder updates, degrading performance by limiting adaptation to temporal and structural patterns.

Table 2: Ablation analysis of key components of ScatterAD across four real-world datasets. Evaluation Metrics: AF (Affiliated-F1), AR (Area Under the ROC Curve). **Bold** indicates the optimal performance; Underline indicates the suboptimal performance.

| Variation | MSL | | PSM | | WADI | | SWaT | | #1 Count |
|---|---|---|---|---|---|---|---|---|---|
| | AF | AR | AF | AR | AF | AR | AF | AR | |
| w/o T-Enc | 0.764 | 0.927 | **0.810** | 0.951 | 0.568 | 0.958 | 0.659 | 0.944 | 1 |
| w/o S-Enc | 0.841 | 0.963 | 0.799 | 0.985 | 0.488 | 0.921 | 0.643 | 0.974 | 0 |
| w/o C-Fuse | 0.795 | 0.932 | 0.788 | 0.946 | 0.549 | 0.953 | 0.701 | 0.975 | 0 |
| w/o EMA | 0.805 | **0.986** | 0.792 | 0.979 | 0.582 | 0.955 | 0.683 | 0.957 | 1 |
| **ScatterAD** | **0.867** | 0.983 | 0.794 | **0.986** | **0.587** | **0.960** | **0.704** | **0.982** | 6 |

## 4.4 Model Analysis

**Analysis of Scattering Mechanism** To validate our hypothesis that dynamically balancing scattering and structure consistency enables learning node representations with both discriminability and temporal dependency, we analyze the evolution of scattering during training (Figure 4). When using only the time consistency loss $\mathcal{L}_{\text{time}}$4a, the heat map shows smooth but low-amplitude scattering, indicating strong temporal continuity but limited anomaly sensitivity. In contrast, using only the scattering loss $\mathcal{L}_{\text{scatter}}$4b increases node dispersion and discriminability but disrupts temporal coherence. Our full method 4c, which combines $\mathcal{L}_{\text{time}}$, $\mathcal{L}_{\text{scatter}}$ and $\mathcal{L}_{\text{contrast}}$, achieves a balanced scattering pattern that preserves structure while enhancing anomaly separability. Compared to single-loss variants, it yields more stable training and more effective representations for temporal anomaly detection.

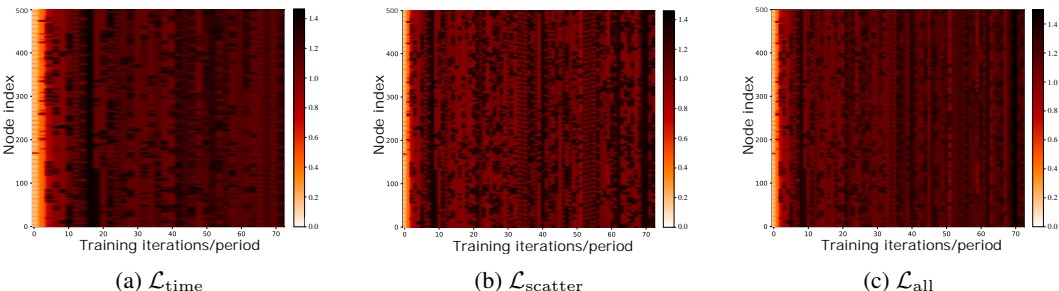

(a) $\mathcal{L}_{\text{time}}$        (b) $\mathcal{L}_{\text{scatter}}$        (c) $\mathcal{L}_{\text{all}}$

Figure 4: Scattering of different time nodes during training on the MSL dataset, measured by the average Euclidean distance between sample representations. (a) Optimized with only $\mathcal{L}_{\text{time}}$. (b) Optimized with only $\mathcal{L}_{\text{scatter}}$. (c) Our method. Darker colors indicate nodes with greater average distances from others. Compared to (a) and (b), our loss function preserves temporal dependencies while promoting more discriminative node representations during training.

**Analysis of Latent Discrepancy in Layer-Wise Evolution** The embedding space transforms significantly across GAT layers, as shown in Figures 5. Initially, normal and anomalous sample embeddings are entangled (Figure 5a). After passing through the GAT layers (Figures 5b and 5c), a structured separation emerges. Normal sample embeddings form cohesive clusters, while anomalous ones scatter outwards. This progression reveals that our model effectively increases the inter-class embedding discrepancy. Rather than explicitly enforcing a margin-based separation, the model implicitly learns to emphasize this discrepancy by capturing richer spatio-temporal interactions through attention mechanisms. This behavior aligns with our notion of spatio-temporal scattering, wherein the joint modeling of temporal dynamics and variable interdependencies causes anomalies to naturally drift away from the normal data manifold in the representation space. Crucially, by amplifying the dissimilarity between normal and anomalous embeddings in both local neighborhoods and global structure, the model enhances its sensitivity to out-of-distribution (OOD) patterns.

**Analysis of Anomaly Criteria** Figure 6 visually compares the anomaly scores of different methods. Our approach demonstrates superior detection across various anomaly types compared to baselines. Specifically, MTGFlow suffers from high false positives, while AnomalyTrans has noticeable false negatives. Sub-Adjacent Trans, despite detecting all anomaly types, also generates significant false

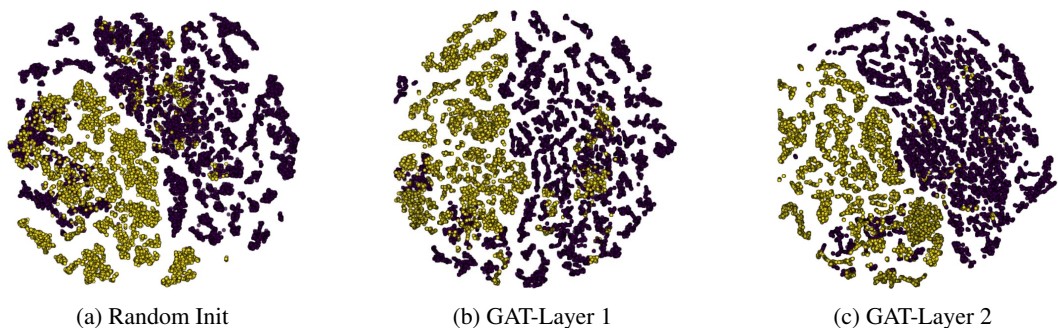

(a) Random Init         (b) GAT-Layer 1         (c) GAT-Layer 2

Figure 5: t-SNE embeddings of GAT on the PSM dataset. Blue and yellow dots denote perturbed negative and positive samples, respectively. The embeddings evolve from an initial entangled state to increasingly disentangled clusters in deeper encoder layers.

alarms, particularly for contextual and shapelet anomalies, where it incorrectly assigns high scores to normal regions. In contrast, our method not only highlights anomalies more effectively but also reduces false positives, leading to improved accuracy and robustness.

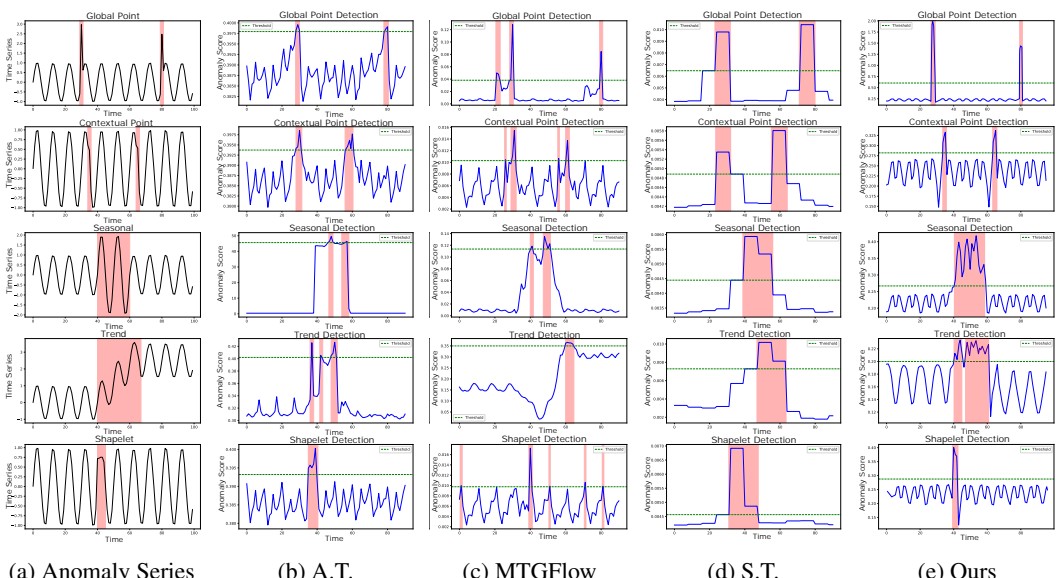

(a) Anomaly Series     (b) A.T.     (c) MTGFlow     (d) S.T.     (e) Ours

Figure 6: Visualization comparison of ground truth anomalies and anomaly scores across different types. The pink segments are marked as anomalies of the pattern.

## 5 Conclusion

In this work, we propose ScatterAD, a novel approach integrating topological and temporal contrastive learning. ScatterAD enhances the discriminability of representations through a scattering mechanism, followed by the application of a topological constraint mechanism to prevent over-scattering and maintain temporal consistency, and introduces a contrastive fusion strategy to foster the synergistic complementarity of temporal and topological representations to improve anomaly detection performance. Comprehensive experimentation demonstrates that ScatterAD attains strong performance across diverse benchmarks. The proposed temporal and topological representation fusion mechanism significantly improves the model's ability to distinguish normal and anomalous patterns, providing valuable insights for developing more effective anomaly detection systems.

# 6 Acknowledgments

This work is supported by the National Key Research and Development Project of China (No. 2024YFB3309900), the Fundamental Research Funds for the Central Universities (No. 2025CDJZKKYJH-08), the National Natural Science Foundation of China (No. 62476101), the Chongqing Technology Innovation and Application Development Project (No. CSTB2023TIAD-STX0025), and the Guangdong Basic and Applied Basic Research Foundation (No. 2024A1515140137).

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

# A  Proof of Theorems

Mutual Information and Representation Learning: In representation learning, we focus on how the representation $Z$ extracted from the original data $X$ retains information related to the target variable $Y$. Mutual information $I(Z; Y)$ measures the amount of information about $Y$ contained in $Z$:

$$I(Z; Y) = \int_z \int_y p(z, y) \log \frac{p(z, y)}{p(z)p(y)} \, dy \, dz. \tag{11}$$

For discrete variables, the above integral becomes a summation. Equivalently,

$$I(Z; Y) = H(Y) - H(Y \mid Z). \tag{12}$$

Representation learning aims to balance information retention and representation complexity:

$$\mathcal{J}(Z) = I(Z; Y) - \lambda \mathcal{C}(Z), \tag{13}$$

where $\mathcal{C}(Z)$ measures complexity and $\lambda > 0$ controls the tradeoff. We consider a temporal encoder $E_T$ and a topological encoder $E_G$:

$$Z_T = E_T(X), \quad Z_G = E_G(X), \quad Z = f(Z_T, Z_G \mid G). \tag{14}$$

Here $f(\cdot \mid G)$ is a graph-guided fusion mechanism. **Assumptions.** We assume the graph $G$ is constructed from domain knowledge and is independent of both the target $Y$ and the temporal representation $Z_T$:

$$I(G; Y) = 0, \quad I(G; Z_T) = 0. \tag{15}$$

**Theorem 1 (Graph-Conditioned Information Bound)**

Let $Z = f(Z_T, Z_G \mid G)$ be the fused representation. Then

$$I(Z; Y) \leq \min\{I(X; Y), \; I(Z_T, Z_G; Y \mid G)\}. \tag{16}$$

Moreover, under the above assumptions, one obtains

$$I(Z; Y) \leq I(Z_T; Y) + I(Z_G; Y \mid Z_T). \tag{17}$$

By the Data Processing Inequality (DPI) and the Markov chain $X \to (Z_T, Z_G, G) \to Z$, we have

$$I(Z; Y) \leq I(X; Y). \tag{18}$$

Similarly,

$$I(Z; Y) \leq I(Z_T, Z_G, G; Y). \tag{19}$$

Applying the chain rule to $I(Z_T, Z_G, G; Y)$ gives

$$I(Z_T, Z_G, G; Y) = I(G; Y) + I(Z_T, Z_G; Y \mid G). \tag{20}$$

Under (15), $I(G; Y) = 0$, hence

$$I(Z; Y) \leq I(Z_T, Z_G; Y \mid G). \tag{21}$$

Next, by chain rule decomposition,

$$I(Z_T, Z_G; Y \mid G) = I(Z_T; Y \mid G) + I(Z_G; Y \mid Z_T, G). \tag{22}$$

By the independence $I(G; Z_T) = 0$ and the fact that conditioning cannot increase mutual information,

$$I(Z_T; Y \mid G) = I(Z_T; Y), \quad I(Z_G; Y \mid Z_T, G) \leq I(Z_G; Y \mid Z_T). \tag{23}$$

Combining (21)–(23) yields (17). Finally, merging with (18) gives the bound in (16).

**Remark.** The contrastive loss in our model maximizes a lower bound on $I(Z_T; Z_G \mid G)$, encouraging complementary information extraction under graph constraints.

**Theorem 2 (Graph-Conditioned Mutual Information Maximization with Asymmetric Encoders)**

Let $Z_T = p_\theta(E_T(X))$ and $Z_G = \mathrm{sg}(E_G(X))$ be the temporal and topological representations under graph $G$, where $p_\theta$ is a predictor and $\mathrm{sg}(\cdot)$ denotes stop-gradient. The contrastive loss:

$$\mathcal{L}_{\mathrm{contrast}} = -\mathbb{E}_{\substack{(s,d)\sim G \\ \mathcal{B}}} \left[ \log \frac{\exp(s_\theta(Z_T[s], Z_G[d]))}{\sum_{d' \in \mathcal{B}} \exp(s_\theta(Z_T[s], Z_G[d']))} \right]. \tag{24}$$

achieves:

$$I(Z_T; Z_G | G) \geq \log(|\mathcal{B}| - 1) - \mathcal{L}_{\mathrm{contrast}}. \tag{25}$$

**Proof:** The proof builds on the variational lower bound for mutual information and its connection to contrastive learning. Variational Lower Bound of Conditional MI. Following the variational framework for mutual information estimation [Oord et al., 2018, Poole et al., 2019], then define conditional mutual information:

$$I(Z_T; Z_G | G) = \mathbb{E}_{(s,d)\sim G} \left[ \mathbb{E}_{Z_T[s], Z_G[d]} \log \frac{p(Z_T[s]; Z_G[d] | G)}{p(Z_T[s] | G)\, p(Z_G[d] | G)} \right]. \tag{26}$$

Introduce variational distribution $q(Z_G[d] | Z_T[s], G)$ to approximate the true conditional distribution. According to the variational lower bound, we get:

$$I(Z_T; Z_G | G) \geq \mathbb{E}_{(s,d)\sim G} \left[ \mathbb{E}_{Z_T[s], Z_G[d]} \log \frac{q(Z_G[d] | Z_T[s], G)}{p(Z_G[d] | G)} \right]. \tag{27}$$

Then, maximize the lower bound by contrast loss:

Assume the scoring function is normalized cosine similarity:

$$s_\theta(Z_T[s], Z_G[d]) = \frac{\langle p_\theta(Z_T[s]), Z_G[d] \rangle}{\tau \, \|p_\theta(Z_T[s])\| \, \|Z_G[d]\|}. \tag{28}$$

Define variational distribution:

$$q(Z_G[d] | Z_T[s], G) = \frac{\exp(s_\theta(Z_T[s], Z_G[d]))}{\sum_{d' \in \mathcal{B}} \exp(s_\theta(Z_T[s], Z_G[d']))}. \tag{29}$$

Substituting into the variational lower bound we get:

$$I(Z_T; Z_G | G) \geq \mathbb{E}_{(s,d)\sim G} \left[ s_\theta(Z_T[s], Z_G[d]) - \log \sum_{d' \in \mathcal{B}} \exp(s_\theta(Z_T[s], Z_G[d'])) \right]. \tag{30}$$

Contrast loss $\mathcal{L}_{\mathrm{contrast}}$ It can be rewritten as:

$$\mathcal{L}_{\mathrm{contrast}} = -\mathbb{E}_{(s,d)\sim G} \left[ s_\theta(Z_T[s], Z_G[d]) \right] + \mathbb{E}_s \left[ \log \sum_{d' \in \mathcal{B}} \exp(s_\theta(Z_T[s], Z_G[d'])) \right]. \tag{31}$$

Combining the two equations, we can get the final lower bound:

$$I(Z_T; Z_G | G) \geq \log(|\mathcal{B}| - 1) - \mathcal{L}_{\mathrm{contrast}}. \tag{32}$$

Therefore, Minimizing the contrastive loss $\mathcal{L}_{\mathrm{contrast}}$ is equivalent to maximizing a lower bound on the conditional mutual information $I(Z_T, Z_G | G)$, and this optimization inherently encourages the temporal embedding $Z_T$ and the topological embedding $Z_G$ to share as much graph-constrained information as possible.

# B Experimental details

## B.1 Datasets

Our model ScatterAD is evaluated on six real world multivariate datasets. The specific descriptions of the datasets are shown in the following table.

Table 3: Dataset statistics used in our experiments. "Dims" denotes the number of dimensions (variables). "Train" and "Test" indicate the number of time points in training and labeled test sets, respectively. "AR" denotes the anomaly rate (%).

| Benchmark | Source | Dims | Train | Test (labeled) | AR (%) |
|---|---|---|---|---|---|
| MSL | NASA Space Sensors | 55 | 58,317 | 73,729 | 10.48 |
| PSM | eBay Server Machine | 25 | 132,481 | 87,941 | 27.76 |
| SWaT | Infrastructure System | 51 | 495,000 | 449,919 | 12.14 |
| WADI | Infrastructure System | 123 | 1,209,601 | 172,801 | 5.71 |
| NIPS-TS-GECCO | Water Quality for IoT | 9 | 69,260 | 69,260 | 1.10 |
| NIPS-TS-SWAN | Solar Weather (Space) | 38 | 60,000 | 60,000 | 32.60 |

The SWaT and WADI datasets can be obtained by filling out the following form: `https://docs.google.com/forms/d/1GOLYXa7TX0KlayqugUOOPMvbcwSQiGNMOjHuNqKcieA/viewform?edit_requested=true`

## B.2 Baselines

For all baseline comparisons, we rely on their official codebases and adopt the hyperparameter settings as suggested in the respective publications. The corresponding source codes are publicly accessible at the URLs provided below.

**Sub-Trans** (IJCAI'24) `https://github.com/jackyue1994/sub_adjacent_transformer`

**ModernTCN** (ICLR'24) `https://github.com/luodhhh/ModernTCN`

**iTransformer** (ICLR'24) `https://github.com/thuml/iTransformer`

**TopoGDN** (CIKM'24) `https://github.com/ljj-cyber/topogdn`

**DCdetector** (KDD'23) `https://github.com/DAMO-DI-ML/KDD2023-DCdetector`

**MEMTO** (NeurIPS'23) `https://github.com/gunny97/MEMTO`

**DuoGAT** (CIKM'23) `https://github.com/ByeongtaePark/DuoGAT`

**MTGFlow** (AAAI'23) `https://github.com/zqhang/MTGFLOW`

**GANF** (ICLR'22) `https://github.com/EnyanDai/GANF`

**AnomalyTrans** (ICLR'22) `https://github.com/thuml/Anomaly-Transformer`

**VAE** `https://github.com/yzhao062/pyod/blob/master/pyod/models/vae.py`

**IF** `https://github.com/yzhao062/pyod/blob/master/pyod/models/iforest.py`

**PCA** `https://github.com/yzhao062/pyod/blob/master/pyod/models/pca.py`

## B.3 Evaluation Metrics

In anomaly detection evaluation, to comprehensively assess model performance across diverse anomaly detection scenarios, we adopt a unified evaluation protocol inspired by Liu et al. [Liu and Paparrizos, 2024], covering both label-based and score-based metrics.

**For label-based evaluation** We use the Affiliated-Precision (Aff-P), Affiliated-Recall (Aff-R), and Affiliated-F1 (Aff-F) [Huet et al., 2022], which aim to better reflect the proximity between predicted and true anomaly ranges. For completeness, we also include the widely used yet imperfect point-adjusted metric, including Point-Adjusted-Precision (PA-P), Point-Adjusted-Recall (PA-R) and Point-Adjusted-F1 (PA-F).

**score-based evaluation** We employ conventional metrics such as (AUC-PR) and (AUC-ROC), along with their range-based extensions, Range-AUC-PR (R-A-P) and Range-AUC-ROC (R-A-R) [Paparrizos et al., 2022], which consider anomaly duration and sequential nature to more comprehensively reflect model performance. Additionally, we incorporate the Volume Under the Surface measures, VUS-ROC(V-ROC) and VUS-PR(V-PR) [Paparrizos et al., 2022], which enhance anomaly score reliability and interpretability by introducing tolerance buffers around anomaly boundaries and using continuous labels.

### B.4 Implementation Details

All experiments were performed in Python 3.9 using PyTorch and on NVIDIA Tesla-A800 GPUs. We used the Adam optimizerduring training. Initially, the batch size was set to 128, and the time window size was uniformly set to 110 for all datasets except NIPS-TS-GECCO(N-T-W) and NIP-TS-Swan(N-T-S), which were set to 90. The GAT in ScatterAD has H=4 attention heads per layer and a hidden dimension of 512. For hyperparameter tuning, the training set was temporarily divided into 80% for training and 20% for validation. For each dataset, the learning rate was uniformly set to 0.0001.

## C   Additional Experiments

Due to space constraints in the main text, we present additional experimental results in this section. In particular, we further evaluate the performance using the following metrics: Range-AUC-ROC and Range-AUC-PR, VUS-ROC and VUS-PR, Affiliated Precision, Affiliated Recall and Affiliated F1, as well as point-adjusted metrics including PA-P, PA-R and PA-F. The reported results are averaged over multiple runs to ensure reliability.

Table 4: Quantitative comparison of four evaluation metrics. R-A-R and R-A-P are Range-AUC-ROC and Range-AUC-PR. V-ROC and V-PR are volumes under the surfaces created based on the ROC curve and PR curve. **Bold** indicates the optimal performance; Underline indicates the suboptimal performance.

| Dataset | Metric | Ours | S.T. | T.G. | Memto | DC. | M.TCN | iT. | MTG | A.T. | D.G. | GANF | VAE | IF | PCA |
|---|---|---|---|---|---|---|---|---|---|---|---|---|---|---|---|
| MSL | R-A-R | **0.920** | 0.750 | 0.695 | 0.704 | 0.913 | 0.756 | 0.655 | 0.748 | 0.891 | 0.613 | 0.846 | 0.725 | 0.641 | 0.633 |
| | R-A-P | **0.919** | 0.754 | 0.826 | 0.643 | 0.891 | 0.745 | 0.638 | 0.753 | 0.875 | 0.574 | 0.733 | 0.630 | 0.654 | 0.651 |
| | V-ROC | 0.872 | 0.745 | 0.705 | 0.601 | **0.904** | 0.748 | 0.647 | 0.648 | 0.884 | 0.512 | 0.644 | 0.626 | 0.743 | 0.641 |
| | V-PR | **0.885** | 0.752 | 0.785 | 0.639 | 0.883 | 0.737 | 0.631 | 0.727 | 0.864 | 0.573 | 0.678 | 0.631 | 0.656 | 0.652 |
| PSM | R-A-R | 0.889 | 0.731 | 0.764 | 0.820 | **0.911** | 0.781 | 0.738 | 0.703 | 0.874 | 0.588 | 0.686 | 0.641 | 0.692 | 0.616 |
| | R-A-P | 0.918 | 0.806 | 0.591 | 0.871 | **0.925** | 0.841 | 0.808 | 0.831 | 0.865 | 0.629 | 0.821 | 0.571 | 0.712 | 0.671 |
| | V-ROC | **0.879** | 0.729 | 0.641 | 0.814 | 0.861 | 0.784 | 0.741 | 0.707 | 0.844 | 0.583 | 0.687 | 0.574 | 0.652 | 0.516 |
| | V-PR | **0.911** | 0.798 | 0.795 | 0.867 | 0.888 | 0.842 | 0.808 | 0.825 | 0.904 | 0.525 | 0.813 | 0.738 | 0.659 | 0.569 |
| SWaT | R-A-R | 0.942 | 0.956 | 0.734 | 0.710 | **0.967** | 0.791 | 0.931 | 0.539 | 0.952 | 0.563 | 0.568 | 0.732 | 0.638 | 0.663 |
| | R-A-P | 0.915 | 0.868 | 0.646 | 0.678 | **0.941** | 0.772 | 0.886 | 0.551 | 0.795 | 0.484 | 0.424 | 0.648 | 0.674 | 0.694 |
| | V-ROC | 0.945 | 0.938 | 0.776 | 0.713 | **0.970** | 0.791 | 0.924 | 0.540 | 0.954 | 0.322 | 0.563 | 0.842 | 0.548 | 0.707 |
| | V-PR | 0.917 | 0.870 | 0.637 | 0.681 | **0.944** | 0.771 | 0.881 | 0.551 | 0.796 | 0.383 | 0.414 | 0.656 | 0.678 | 0.689 |
| WADI | R-A-R | 0.927 | 0.945 | 0.623 | 0.866 | 0.912 | 0.561 | 0.629 | 0.614 | **0.955** | 0.413 | 0.617 | 0.692 | 0.714 | 0.677 |
| | R-A-P | **0.846** | 0.843 | 0.588 | 0.797 | 0.773 | 0.265 | 0.359 | 0.637 | 0.794 | 0.442 | 0.639 | 0.549 | 0.434 | 0.547 |
| | V-ROC | 0.877 | 0.943 | 0.627 | 0.758 | 0.937 | 0.573 | 0.631 | 0.618 | **0.956** | 0.412 | 0.620 | 0.692 | 0.719 | 0.669 |
| | V-PR | 0.799 | **0.854** | 0.576 | 0.659 | 0.786 | 0.276 | 0.359 | 0.634 | 0.796 | 0.441 | 0.634 | 0.549 | 0.439 | 0.542 |
| NIPS-TS-SWAN | R-A-R | 0.858 | 0.870 | 0.607 | **0.882** | 0.881 | 0.872 | 0.866 | 0.843 | 0.878 | 0.729 | 0.748 | 0.807 | 0.702 | 0.794 |
| | R-A-P | 0.941 | 0.936 | 0.587 | 0.851 | **0.948** | 0.946 | 0.944 | 0.879 | 0.929 | 0.705 | 0.781 | 0.639 | 0.646 | 0.743 |
| | V-ROC | 0.829 | 0.851 | 0.607 | **0.873** | 0.861 | 0.853 | 0.847 | 0.805 | 0.866 | 0.607 | 0.807 | 0.755 | 0.691 | 0.681 |
| | V-PR | 0.932 | 0.925 | 0.587 | 0.929 | **0.935** | 0.933 | 0.931 | 0.858 | 0.918 | 0.687 | 0.759 | 0.629 | 0.637 | 0.633 |
| NIPS-TS-GECCO | R-A-R | **0.721** | 0.536 | 0.599 | 0.523 | 0.629 | 0.522 | 0.523 | 0.566 | 0.581 | 0.522 | 0.564 | 0.614 | 0.635 | 0.626 |
| | R-A-P | **0.587** | 0.318 | 0.585 | 0.483 | 0.345 | 0.311 | 0.367 | 0.566 | 0.349 | 0.519 | 0.526 | 0.561 | 0.522 | 0.517 |
| | V-ROC | **0.748** | 0.541 | 0.589 | 0.527 | 0.623 | 0.521 | 0.521 | 0.568 | 0.576 | 0.513 | 0.565 | 0.669 | 0.665 | 0.624 |
| | V-PR | **0.611** | 0.323 | 0.582 | 0.479 | 0.339 | 0.314 | 0.353 | 0.546 | 0.443 | 0.512 | 0.505 | 0.597 | 0.582 | 0.554 |
| #1 Count | | **12** | 1 | 0 | 2 | 9 | 0 | 0 | 0 | 2 | 0 | 0 | 0 | 0 | 0 |

Table 5: Performance comparison across six benchmark datasets using affinity-based precision (Aff-P), recall (Aff-R), and F1 score (Aff-F). Our method consistently outperforms others, especially in Aff-F. **Bold** indicates the optimal performance; Underline indicates the suboptimal performance.

| Dataset | Metric | Ours | S.T. | T.G. | Memto | DC. | M.TCN | iT. | MTG | A.T. | D.G. | GANF | VAE | IF | PCA |
|---|---|---|---|---|---|---|---|---|---|---|---|---|---|---|---|
| MSL | Aff-P | **0.994** | 0.549 | 0.709 | 0.515 | 0.516 | 0.588 | 0.559 | 0.921 | 0.518 | 0.512 | 0.762 | 0.587 | 0.423 | 0.694 |
| | Aff-R | 0.782 | 0.871 | 0.642 | 0.704 | 0.972 | 0.893 | 0.785 | 0.234 | 0.961 | **0.998** | 0.205 | 0.714 | 0.342 | 0.512 |
| | Aff-F | **0.867** | 0.673 | 0.674 | 0.595 | 0.674 | 0.709 | 0.652 | 0.374 | 0.673 | 0.677 | 0.323 | 0.642 | 0.374 | 0.591 |
| PSM | Aff-P | 0.746 | 0.774 | 0.627 | 0.641 | 0.561 | 0.664 | 0.644 | **0.967** | 0.551 | 0.465 | 0.961 | 0.755 | 0.469 | 0.389 |
| | Aff-R | 0.846 | 0.798 | 0.764 | 0.677 | 0.783 | 0.743 | 0.662 | 0.282 | 0.814 | **0.945** | 0.198 | 0.412 | 0.731 | 0.312 |
| | Aff-F | **0.797** | 0.786 | 0.689 | 0.659 | 0.653 | 0.701 | 0.652 | 0.436 | 0.657 | 0.624 | 0.329 | 0.524 | 0.569 | 0.437 |
| SWaT | Aff-P | 0.567 | 0.582 | 0.538 | 0.554 | 0.531 | 0.586 | 0.581 | 0.662 | 0.571 | 0.478 | 0.552 | 0.499 | **0.743** | 0.387 |
| | Aff-R | 0.926 | 0.689 | 0.663 | 0.635 | **0.981** | 0.827 | 0.934 | 0.356 | 0.683 | 0.878 | 0.676 | 0.642 | 0.363 | 0.725 |
| | Aff-F | 0.704 | 0.631 | 0.594 | 0.592 | 0.687 | 0.685 | **0.716** | 0.463 | 0.622 | 0.619 | 0.608 | 0.563 | 0.502 | 0.537 |
| WADI | Aff-P | 0.714 | 0.632 | 0.652 | 0.545 | 0.619 | 0.546 | 0.573 | **0.771** | 0.553 | 0.476 | 0.751 | 0.456 | 0.658 | 0.601 |
| | Aff-R | 0.525 | 0.956 | 0.449 | 0.981 | 0.874 | 0.571 | 0.809 | 0.598 | **0.985** | 0.957 | 0.601 | 0.726 | 0.482 | 0.365 |
| | Aff-F | 0.605 | **0.756** | 0.532 | 0.701 | 0.725 | 0.558 | 0.671 | 0.673 | 0.708 | 0.636 | 0.667 | 0.556 | 0.555 | 0.477 |
| NIPS-TS-SWAN | Aff-P | 0.754 | 0.798 | 0.521 | 0.525 | 0.531 | 0.515 | 0.491 | **0.998** | 0.672 | 0.509 | 0.435 | 0.374 | 0.711 | 0.645 |
| | Aff-R | 0.019 | 0.824 | **0.997** | 0.017 | 0.445 | 0.554 | 0.526 | 0.002 | 0.054 | 0.935 | 0.003 | 0.396 | 0.318 | 0.731 |
| | Aff-F | 0.038 | **0.805** | 0.685 | 0.033 | 0.484 | 0.533 | 0.507 | 0.003 | 0.099 | 0.659 | 0.005 | 0.385 | 0.438 | 0.680 |
| NIPS-TS-GECCO | Aff-P | **0.829** | 0.805 | 0.508 | 0.638 | 0.513 | 0.555 | 0.541 | 0.706 | 0.533 | 0.501 | 0.592 | 0.732 | 0.402 | 0.539 |
| | Aff-R | 0.820 | 0.337 | 0.702 | 0.317 | 0.883 | 0.373 | 0.365 | 0.231 | 0.859 | **0.998** | 0.173 | 0.362 | 0.698 | 0.481 |
| | Aff-F | **0.825** | 0.476 | 0.589 | 0.424 | 0.648 | 0.446 | 0.435 | 0.348 | 0.658 | 0.667 | 0.268 | 0.481 | 0.531 | 0.509 |
| #1 Count | | **6** | 2 | 1 | 0 | 1 | 0 | 1 | 3 | 1 | 3 | 0 | 0 | 1 | 0 |

Table 6: Performance comparison in terms of PA-Precision (PA-P), PA-Recall (PA-R), and PA-F1 (PA-F) on six benchmark datasets. **Bold** indicates the optimal performance; Underline indicates the suboptimal performance.

| Dataset | Metric | Ours | S.T. | T.G. | Memto | DC. | M.TCN | iT. | MTG | A.T. | D.G. | GANF | VAE | IF | PCA |
|---|---|---|---|---|---|---|---|---|---|---|---|---|---|---|---|
| MSL | PA-P | 0.933 | 0.931 | 0.626 | **0.971** | 0.922 | 0.884 | 0.825 | 0.744 | 0.917 | 0.742 | 0.422 | 0.421 | 0.509 | 0.472 |
| | PA-R | **0.997** | 0.803 | 0.836 | 0.504 | 0.974 | 0.743 | 0.548 | 0.574 | 0.951 | 0.582 | 0.584 | 0.635 | 0.723 | 0.416 |
| | PA-F | **0.964** | 0.863 | 0.714 | 0.664 | 0.947 | 0.807 | 0.659 | 0.648 | 0.934 | 0.652 | 0.489 | 0.512 | 0.598 | 0.444 |
| PSM | PA-P | 0.985 | 0.980 | 0.784 | 0.989 | 0.971 | 0.975 | 0.962 | 0.992 | 0.973 | 0.837 | **0.994** | 0.691 | 0.374 | 0.647 |
| | PA-R | 0.977 | 0.904 | 0.819 | 0.966 | 0.984 | 0.954 | 0.893 | 0.666 | **0.983** | 0.731 | 0.652 | 0.524 | 0.739 | 0.337 |
| | PA-F | **0.981** | 0.941 | 0.801 | 0.977 | 0.977 | 0.965 | 0.926 | 0.797 | 0.978 | 0.780 | 0.788 | 0.596 | 0.534 | 0.467 |
| SWaT | PA-P | 0.931 | 0.809 | 0.827 | 0.889 | 0.931 | 0.894 | 0.874 | **0.952** | 0.892 | 0.786 | 0.548 | 0.585 | 0.651 | 0.443 |
| | PA-R | 0.973 | 0.923 | 0.754 | 0.659 | 0.949 | 0.875 | 0.962 | 0.338 | **0.992** | 0.863 | 0.157 | 0.477 | 0.328 | 0.721 |
| | PA-F | **0.951** | 0.863 | 0.788 | 0.756 | 0.939 | 0.884 | 0.916 | 0.500 | 0.943 | 0.822 | 0.244 | 0.526 | 0.472 | 0.548 |
| WADI | PA-P | 0.813 | 0.856 | 0.841 | 0.751 | 0.607 | 0.291 | 0.355 | **0.875** | 0.637 | 0.866 | 0.847 | 0.658 | 0.372 | 0.547 |
| | PA-R | 0.917 | 0.890 | 0.829 | 0.874 | 0.917 | 0.372 | 0.511 | 0.570 | **0.947** | 0.595 | 0.570 | 0.416 | 0.612 | 0.723 |
| | PA-F | 0.862 | **0.873** | 0.834 | 0.807 | 0.731 | 0.326 | 0.419 | 0.690 | 0.738 | 0.705 | 0.681 | 0.512 | 0.474 | 0.618 |
| NIPS-TS-SWAN | PA-P | 0.991 | **0.997** | 0.694 | 0.988 | 0.966 | 0.954 | 0.952 | 0.662 | 0.972 | 0.591 | 0.695 | 0.633 | 0.364 | 0.699 |
| | PA-R | 0.585 | 0.556 | 0.629 | 0.526 | 0.591 | 0.592 | 0.591 | 0.533 | 0.540 | 0.438 | 0.576 | 0.395 | **0.738** | 0.403 |
| | PA-F | **0.736** | 0.714 | 0.660 | 0.687 | 0.733 | 0.731 | 0.729 | 0.591 | 0.695 | 0.503 | 0.630 | 0.497 | 0.522 | 0.526 |
| NIPS-TS-GECCO | PA-P | 0.669 | 0.524 | 0.582 | **0.889** | 0.391 | 0.404 | 0.471 | 0.371 | 0.277 | 0.539 | 0.346 | 0.382 | 0.548 | 0.466 |
| | PA-R | **0.944** | 0.463 | 0.789 | 0.352 | 0.597 | 0.321 | 0.321 | 0.391 | 0.391 | 0.745 | 0.364 | 0.672 | 0.422 | 0.735 |
| | PA-F | **0.784** | 0.491 | 0.670 | 0.504 | 0.472 | 0.357 | 0.381 | 0.381 | 0.325 | 0.625 | 0.355 | 0.491 | 0.481 | 0.583 |
| #1 Count | | **6** | 2 | 0 | 2 | 0 | 0 | 0 | 2 | 3 | 0 | 1 | 0 | 1 | 0 |

## D  Analysis on the Universality and Robustness of the Scattering Phenomenon

To investigate the universality of the scattering phenomenon and evaluate the model's robustness, we conduct a quantitative analysis across all six benchmark datasets under varying levels of noise. Specifically, we inject additive Gaussian noise with four different intensities ($\sigma \in \{0, 0.5, 1.0, 2.0\}$) into the normalized test data, where $\sigma = 0.0$ serves as the no-noise baseline. We then quantify the model's internal discriminative power by calculating the average scattering score for both normal and anomalous samples. This score is defined as the mean Euclidean distance of samples to their respective class center. The separation ratio between these scores (Anomalous/Normal) is analyzed to measure class distinguishability, with results presented in Table 7.

Table 7: Analysis of scattering scores and their separation ratio under different Gaussian noise levels ($\sigma$). A separation ratio greater than 1.0 suggests that anomalous samples are, on average, farther from their class center than the normal samples, confirming the scattering phenomenon.

| Dataset | Noise Level ($\sigma$) | Scattering Score (Normal) | Scattering Score (Anomalous) | Separation Ratio(%) |
|---|---|---|---|---|
| MSL | 0.0 | 16.0 | 18.2 | 1.14 |
| | 0.5 | 21.6 | 31.6 | 1.46 |
| | 1.0 | 36.0 | 49.3 | 1.37 |
| | 2.0 | 65.9 | 84.6 | 1.28 |
| PSM | 0.0 | 31.5 | 37.8 | 1.20 |
| | 0.5 | 33.1 | 40.3 | 1.22 |
| | 1.0 | 41.2 | 47.4 | 1.15 |
| | 2.0 | 61.1 | 65.7 | 1.08 |
| SWaT | 0.0 | 73.10 | 94.09 | 1.29 |
| | 0.5 | 75.63 | 95.71 | 1.27 |
| | 1.0 | 79.22 | 98.03 | 1.24 |
| | 2.0 | 84.50 | 102.11 | 1.21 |
| WADI | 0.0 | 0.78 | 1.16 | 1.49 |
| | 0.5 | 34.9 | 43.4 | 1.24 |
| | 1.0 | 87.1 | 72.4 | 0.83 |
| | 2.0 | 166.7 | 137.0 | 0.82 |
| NIPS-TS-SWAN | 0.0 | 20.2 | 28.4 | 1.41 |
| | 0.5 | 21.8 | 32.4 | 1.49 |
| | 1.0 | 27.7 | 42.2 | 1.52 |
| | 2.0 | 42.1 | 65.5 | 1.56 |
| NIPS-TS-GECCO | 0.0 | 21.60 | 95.97 | 4.44 |
| | 0.5 | 26.10 | 105.33 | 4.04 |
| | 1.0 | 40.70 | 41.40 | 1.02 |
| | 2.0 | 68.60 | 197.30 | 2.88 |

The results presented in Table 7 clearly reveal a consistent anomaly scattering pattern across the six diverse, real-world datasets. Notably, for the WADI dataset, the scattering score of normal samples surpasses that of anomalies under medium-to-high noise conditions ($\sigma \geq 1.0$), resulting in a separation ratio less than 1.0. We attribute this phenomenon to WADI's high dimensionality (123 dimensions), where strong noise may have a compounding effect and lead to complex changes in inter-variable correlations. This finding indicates that while our method is generally robust across various noisy environments, its performance may be sensitive to strong noise when applied to extremely high-dimensional datasets that feature subtle anomaly patterns. This analysis helps to comprehensively characterize the performance boundaries of our method.

## E  Parameter Sensitivity Analysis

We analyze the sensitivity of ScatterAD to key hyperparameters across six datasets (Figure 7). The model performs consistently across varying window sizes, demonstrating robustness to both short- and long-range temporal contexts. A window size of 110 yields the highest AUC-ROC and is thus

used in all experiments. Therefore, this brief window size is selected in our experiments. We further studied the sensitivity of the other 2 parameters: the dimension of the hidden layer and the number of encoder layers. For encoder depth, two layers consistently achieve the best performance while keeping the model lightweight. Increasing the hidden dimension improves performance, and we choose 512 as a balanced default to maintain model efficiency.

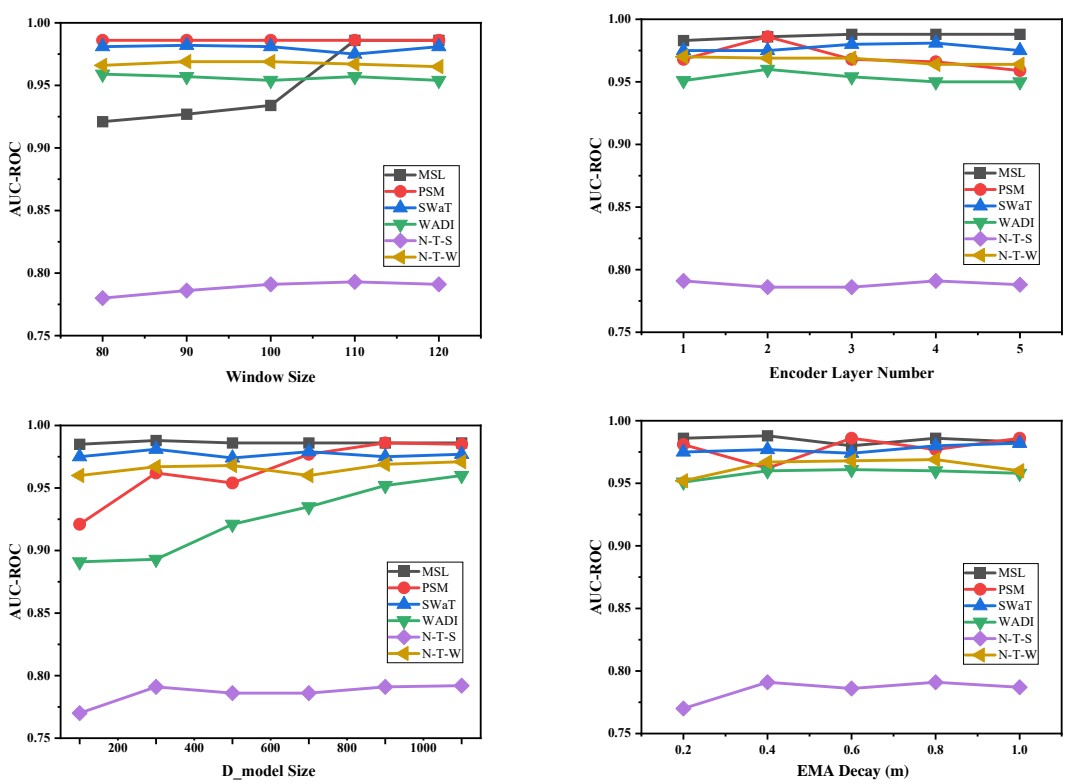

Figure 7: Parameter sensitivity analysis in ScatterAD.

Anomaly threshold $\sigma$ is a important hyperparameter, which may affect the determination of anomaly or not. We have a default value of 1 for all benchmarks. As shown in Table 8, when it is in the range of 0.5 to 1, it has little effect on the final model performance. PSM and SMAP are also more robust to anomaly thresholds than MSL. For the three benchmarks, its best results appear when $\sigma$ equals 0.8 or 1.

Table 8: Sensitivity analysis of the hyperparameter $\delta$ on six datasets. Metrics: AF (Affiliated-F1), AR (AUC-ROC). Best values per column are highlighted in bold.

| Dataset | MSL | | PSM | | SWaT | | WADI | | N-T-S | | N-T-W | |
|---|---|---|---|---|---|---|---|---|---|---|---|---|
| Metric | AF | AR | AF | AR | AF | AR | AF | AR | AF | AR | AF | AR |
| $\delta = 0.2$ | 0.863 | 0.986 | 0.789 | 0.955 | 0.656 | 0.517 | 0.929 | 0.944 | 0.029 | 0.786 | 0.799 | 0.954 |
| $\delta = 0.4$ | 0.861 | 0.982 | 0.793 | 0.952 | 0.698 | 0.521 | 0.934 | 0.974 | 0.038 | 0.792 | 0.803 | 0.964 |
| $\delta = 0.6$ | 0.843 | 0.978 | 0.783 | 0.946 | **0.704** | 0.586 | **0.961** | **0.975** | 0.035 | 0.790 | 0.814 | 0.965 |
| $\delta = 0.8$ | 0.855 | 0.981 | **0.794** | **0.969** | 0.701 | **0.587** | 0.960 | 0.957 | **0.038** | **0.792** | 0.818 | 0.959 |
| $\delta = 1.0$ | **0.867** | **0.986** | 0.792 | **0.969** | 0.702 | 0.583 | 0.955 | 0.957 | 0.037 | 0.789 | **0.823** | **0.969** |

# F Analysis of Performance on the NIPS-TS-SWAN Dataset

A notable performance discrepancy is observed on the NIPS-TS-SWAN dataset, specifically concerning the Affiliated-F1 (Aff-F1) metric. It is important to first establish that our proposed model, ScatterAD, successfully generates separable anomaly scores for this dataset. The high Point-Adjusted F1 (PA-F1) score of 0.736 (as reported in Table 1) demonstrates that anomalous events are correctly assigned higher scores than normal ones at the point level.

To investigate the low Aff-F1 score, we hypothesize that the metric itself may be overly sensitive to minor localization errors on this particular dataset. To test this, we devised a controlled simulation experiment, independent of our model, to evaluate the intrinsic robustness of the Aff-F1 metric. In this simulation, we assume a perfect model whose predictions exactly match the ground truth (achieving an F1 score of 1.0). We then introduce a minor, realistic localization error by shifting the entire prediction sequence by a small number of timesteps ($\Delta t$) and observe the resulting changes in both Aff-F1 and a traditional point-wise F1 score. The experiment is also conducted on the MSL dataset for comparison.

Table 9: Sensitivity analysis of F1 metrics to temporal localization errors. A perfect prediction sequence is shifted by a few timesteps, and the degradation of segment-level (Aff-F1) and point-level F1 scores is measured.

| Dataset | Localization Error (Shift in steps) | Aff-F1 Score (Segment-level) | Point-wise F1 (Point-level) |
|---|---|---|---|
| NIPS-TS-SWAN | 0 steps | 1.000 | 1.000 |
| | 1 step | 0.065 (93.5%↓) | 0.686 |
| | 2 steps | 0.174 | 0.697 |
| | 5 steps | 0.223 | 0.704 |
| | 10 steps | 0.226 | 0.709 |
| MSL | 0 steps | 1.000 | 1.000 |
| | 1 step | 1.000 | 0.995 |
| | 2 steps | 1.000 | 0.991 |
| | 5 steps | 0.972 (2.8%↓) | 0.977 |
| | 10 steps | 0.889 | 0.954 |

The results in Table 9 are revealing. For the NIPS-TS-SWAN dataset, a minuscule localization shift of just one timestep causes the Aff-F1 score to plummet from 1.0 to 0.065, a 93.5% reduction. In stark contrast, the same shift on the MSL dataset has no impact on the Aff-F1 score, and even a 5-step shift results in only a minor 2.8% drop. This demonstrates the extreme sensitivity of the Aff-F1 metric specifically on the NIPS-TS-SWAN dataset. This inherent characteristic of the metric and dataset interaction explains why our model, along with many other state-of-the-art methods (as shown in Table 1), exhibits poor performance on this particular metric. Conversely, the more traditional point-wise F1 score shows much greater resilience. This justifies our decision to report both scores for a balanced and comprehensive evaluation.

To further validate our model's detection capability at a macro level, we compiled overall prediction statistics for ScatterAD on the NIPS-TS-SWAN dataset, shown in Table 10.

Table 10: Macro-level prediction statistics for ScatterAD on the NIPS-TS-SWAN dataset.

| Statistic | Ground Truth | Our Prediction |
|---|---|---|
| Total Anomaly Ratio | 32.6% | 32.1% |
| Number of Anomaly Segments | 551,937 | 843,596 |

The statistics show that the total proportion of anomalies predicted by ScatterAD (32.1%) closely aligns with the ground truth (32.6%), demonstrating its accurate high-level detection capability. However, the model identifies a higher number of individual anomaly segments. This suggests that the

model may tend to partition a single continuous ground-truth anomaly into several smaller, consecutive segments. Consequently, due to such minor, unavoidable localization offsets, its performance is significantly underestimated by the strict segment-matching criteria of the Aff-F1 metric under these specific data conditions.

# G   Analysis of the Scattering Center Initialization

The initialization strategy for the scattering center, $\mathbf{c}$, is a key design consideration impacting model stability and the potential for learning suboptimal solutions. In our framework, the scattering center $\mathbf{c}$ is initialized once at the beginning of each training run and remains fixed throughout the training process; it is not a learnable parameter updated via backpropagation. The role of this fixed center is twofold:

- **Provide a Fixed Anchor Point:** The center $\mathbf{c}$ offers a stable, convergent target direction for the representations of normal samples. This guides the model to learn a mapping that projects these representations into a compact region on the latent hypersphere.

- **Break Symmetry and Regularize:** A fixed, predefined center (e.g., a canonical basis vector like [1, 0, ..., 0]) might introduce an undesirable bias, potentially inducing the model to learn a trivial solution where all normal samples map to a specific axis. By randomly initializing $\mathbf{c}$, we avoid this risk and compel the model to learn more generalizable, rotation-invariant representations. This can be viewed as a form of implicit stochastic regularization, preventing the model from overfitting to a predefined direction in the latent space.

## G.1   Stability Analysis

To empirically validate the stability of this random initialization strategy, we conducted an experiment by repeating the full training and evaluation process with 5 different random seeds on the MSL and PSM datasets. A different seed results in a different initialization for the scattering center $\mathbf{c}$. As shown in Table 11, the standard deviation of the performance metrics is extremely low, confirming that the model's performance is robust and not sensitive to the specific random choice of the center.

Table 11: Stability analysis of the random scattering center initialization across 5 runs with different random seeds. The low standard deviation indicates high stability.

| Dataset | Metric | Original | Mean | Std. Dev. |
|---------|--------|----------|------|-----------|
| MSL | Aff-F1 | 0.865 | 0.865 | $\pm$ 0.003 |
|     | A-ROC  | 0.986 | 0.986 | $\pm$ 0.001 |
| PSM | Aff-F1 | 0.797 | 0.797 | $\pm$ 0.002 |
|     | A-ROC  | 0.986 | 0.986 | $\pm$ 0.003 |

## G.2   Ablation Study on Initialization Strategies

Furthermore, to demonstrate the effectiveness of our chosen approach, we performed an ablation study comparing several alternative center initialization strategies:

- **Zero:** Initialize the center at the origin (zero vector) to test the simplest symmetric starting point.

- **Fixed-Radius:** Initialize the center on a hypersphere with a fixed radius to test the model's sensitivity to the initial magnitude.

- **Multi-center:** Use multiple independent scattering centers to test the model's ability to capture potentially multi-modal distributions of normal data.

The results, presented in Table 12, compare these strategies with the `random_in_ball` approach (randomly selecting a point within the unit hypersphere) proposed in our paper. The findings confirm

that our model is not sensitive to the specific initialization strategy of **c**. The performance remains consistently high across different methods, indicating that the `random_in_ball` strategy is a simple, robust, and effective choice that avoids the need for additional hyperparameter tuning (e.g., setting a radius or the number of centers).

Table 12: Ablation study of different scattering center initialization strategies. Performance metrics (Aff-F1 and AUC ROC) remain stable across various strategies, validating the robustness of our approach.

| Dataset | Center Strategy | Num Centers | Radius | Aff-F1 | AUC-ROC |
|---|---|---|---|---|---|
| MSL | `random_in_ball` (Ours) | 1 | N/A | 0.867 | 0.986 |
| | `zero` | 1 | 0.0 | 0.866 | 0.986 |
| | `fixed_radius` | 1 | 0.3 | 0.866 | 0.986 |
| | `fixed_radius` | 1 | 0.7 | 0.864 | 0.986 |
| | `multi-center` | 3 | N/A | 0.869 | 0.986 |
| PSM | `random_in_ball` (Ours) | 1 | N/A | 0.797 | 0.986 |
| | `zero` | 1 | 0.0 | 0.797 | 0.986 |
| | `fixed_radius` | 1 | 0.3 | 0.797 | 0.986 |
| | `fixed_radius` | 1 | 0.7 | 0.796 | 0.986 |
| | `multi-center` | 3 | N/A | 0.792 | 0.980 |
| SWaT | `random_in_ball` (Ours) | 1 | N/A | 0.704 | 0.982 |
| | `zero` | 1 | 0.0 | 0.704 | 0.982 |
| | `fixed_radius` | 1 | 0.3 | 0.698 | 0.977 |
| | `fixed_radius` | 1 | 0.7 | 0.702 | 0.979 |
| | `multi-center` | 3 | N/A | 0.702 | 0.980 |

## H    Robustness to Graph Topology

To evaluate the model's sensitivity and robustness to the graph structure, we conducted an experiment to simulate dynamically changing graph topologies. We extended our data loading process to generate a different graph structure for each input time window during both training and inference. This contrasts with our main approach of using a single, static graph learned from the training data. We designed two dynamic topology generation strategies:

- **Random Topology:** For each time window, connections between nodes are randomly established with a fixed probability (`edge_prob=0.3`). To ensure basic graph connectivity, the connections between physically adjacent nodes are always preserved. This strategy simulates scenarios where inter-sensor correlations are irregular and appear randomly over time.
- **K-Nearest Neighbors (KNN) Topology:** For each time window, a topology is constructed by computing the Euclidean distance between the time series of each node and all other nodes within that window. Each node is then connected to its $k$-nearest neighbors (we use $k = 3$).

The performance and efficiency of these dynamic strategies were compared against our original static graph approach on the MSL and PSM datasets. The results are presented in Table 13.

The results in Table 13 indicate that the model's detection performance is highly robust to the underlying graph structure. While dynamic topology strategies lead to marginal performance improvements in some specific cases (e.g., Random Topology on MSL and KNN Topology on PSM), the overall accuracy metrics remain remarkably stable across all three approaches. However, these dynamic strategies introduce significant computational overhead, as evidenced by the substantially lower throughput and higher average inference times. This analysis demonstrates that our proposed model

Table 13: Performance and efficiency comparison between static (Original) and dynamic graph topology strategies. While dynamic strategies offer marginal performance changes, they incur significant computational overhead, reducing throughput and increasing inference time.

| Dataset | Strategy | Aff-F | PA-F | A-ROC | A-PR | Throughput (samples/sec) | Avg. Inference Time (ms) |
|---|---|---|---|---|---|---|---|
| MSL | Original (Ours) | 0.866 | 0.964 | 0.986 | 0.932 | 60.35 | 16.57 |
| | Random Topology | 0.867 | 0.967 | 0.987 | 0.937 | 38.33 | 26.09 |
| | KNN Topology | 0.862 | 0.961 | 0.984 | 0.927 | 49.17 | 20.34 |
| PSM | Original (Ours) | 0.797 | 0.981 | 0.986 | 0.969 | 62.17 | 16.09 |
| | Random Topology | 0.791 | 0.975 | 0.980 | 0.961 | 31.77 | 31.47 |
| | KNN Topology | 0.803 | 0.966 | 0.971 | 0.948 | 25.66 | 38.98 |

exhibits strong robustness to graph topology, and employing a static graph derived from the training data strikes an excellent balance between detection effectiveness and computational efficiency.

# I    Convergence analysis

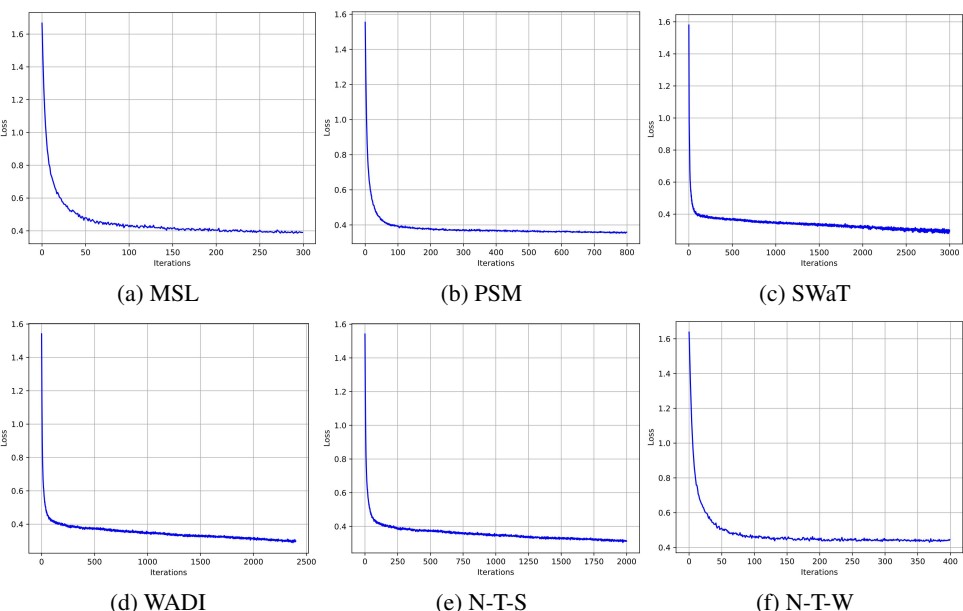

(a) MSL        (b) PSM        (c) SWaT

(d) WADI        (e) N-T-S        (f) N-T-W

Figure 8: The convergence performance of our model's loss on six different datasets is shown in the figure.

The convergence performance of the total loss (Equation 9) of our model on six different datasets is shown in the figure. Several consistent features can be observed from these curves: In all experiments, the loss function shows a similar convergence pattern, rapidly decreasing from an initial value of about 1.6 and gradually stabilizing. Specifically, within the first 100 iterations, the loss value drops rapidly to about 0.4, indicating that the model can effectively capture the key features of the data. Subsequently, the loss curve enters a gentle decline phase and converges to a stable value of about 0.3-0.4. It is worth noting that although the total number of iterations in each experiment is different (300, 800, 2000 and 3000 respectively), they all show similar convergence curve shapes, proving that the temporal topology constraint model we proposed has stable optimization characteristics and can effectively learn the main patterns of the data in a small number of iterations (about 100-200 iterations). This consistent convergence behavior further verifies the robustness and scalability of our model on different datasets.

## J    Run Times

To evaluate the practicality of ScatterAD in production environments, we comprehensively compared the training time, inference time, and GPU memory usage of various neural network models on the SWaT dataset. The results are shown in the table.

Table 14: Comparison of training time, inference time, and GPU memory consumption on the SWaT dataset.

| Method | Training Time (s/iter) | Inference Time (s) | GPU Memory (GB) |
|---|---|---|---|
| GANF | 0.177 | 0.834 | 8.202 |
| DuoGAT | 0.283 | 239.608 | 6.050 |
| AnomalyTrans | 0.378 | 86.134 | 12.424 |
| MTGFlow | 0.378 | 2.119 | 9.458 |
| iTransformer | 0.027 | 59.515 | 2.061 |
| ModernTCN | 0.012 | 59.882 | 1.923 |
| DCdetector | 0.012 | 59.882 | 1.923 |
| MEMTO | 0.621 | 8.582 | 19.91 |
| TopoGDN | 0.131 | 66.146 | 2.830 |
| Sub-Adjacent Trans | 0.107 | 1.762 | 5.396 |
| **ScatterAD** | 0.042 | 2.124 | 3.458 |

Table 14 reports the training and inference time per iteration, along with GPU memory usage on the SWaT dataset. Our method achieves the fastest training speed (**0.0421s/iter**). It also offers competitive inference efficiency and low GPU memory consumption. Compared to DuoGAT and Anomaly Trans, which suffer from extremely long inference times and high memory overhead, our model demonstrates superior scalability and deployment practicality.

## K    Limitations and Future Work

While ScatterAD shows strong performance across diverse real-world datasets, there remain a few avenues for future improvement. ScatterAD effectively captures spatial-temporal dependencies, and incorporating more explicit modeling of inter-series interactions may offer additional benefits in specialized applications such as industrial monitoring or financial forecasting. For future work, we aim to explore adaptive graph construction that evolves with streaming data to improve robustness in non-stationary environments. In addition, integrating causal discovery modules could enhance interpretability and detection precision. Another promising direction is extending ScatterAD to a semi-supervised or active learning setting, where limited anomaly labels can further guide representation learning.

