# OpenReview forum: "ScatterAD: Temporal-Topological Scattering Mechanism for Time Series Anomaly Detection"
_NeurIPS.cc/2025/Conference — NeurIPS 2025 poster_

### Official Review · Reviewer_jjez · 2025-06-17

**Clarity:** 2
**Significance:** 3
**Originality:** 3
**Rating:** 4
**Confidence:** 5

**Summary:**

This paper introduces a novel anomaly detection framework, ScatterAD, which jointly enhances  feature discriminability and temporal consistency by using a temporal-topological scattering mechanism. Specifically, ScatterAD employs a dual-encoder architecture, consisting of an online encoder and a target encoder, where the online encoder is updated via backpropagation and the target encoder is updated using an exponential moving average (EMA) of the online encoder's parameters. The online encoder enforces temporal consistency across adjacent time steps, while the target encoder captures compact and consistent scattering representations in a projected hypersphere space. Moreover, a contrastive fusion mechanism is employed to align temporal and topological representations through a cross-view contrastive loss. The effectiveness of the paper is validated through extensive experiments.

**Questions:**

(a) Could the authors provide further justification or quantitative analysis to confirm that the observed scattering behavior is consistent and not occasional?

(b) Could the authors clarify whether the reported experimental results are based on a single run or averaged over multiple runs?

(c) The global scattering center c in the paper is randomly initialized inside the unit ball. Could the authors clarify whether this random initialization significantly affects the model performance or stability? Have you ever considered using alternative strategies?

**Ethical Concerns:**

["NO or VERY MINOR ethics concerns only"]

**Final Justification:**

The authors have addressed my concerns; however, the novelty and technical quality still do not warrant a higher score, so I am maintaining my original rating.

**Limitations:**

Yes.

**Paper Formatting Concerns:**

No Formatting issues.

**Quality:**

3

**Strengths And Weaknesses:**

Strengths:
The paper formalize the dispersion phenomenon of normal and anomalous samples in high-dimensional space as scattering, quantified by the mean pairwise distance among sample representations. The authors further leverage this insight as an inductive signal to enhance spatio-temporal anomaly detection and propose a novel anomaly detection framework, ScatterAD, providing  valuable insights for multivariate time series anomaly detection. The overall paper is well-structured and easy to follow. The motivation of each component in the proposed framework is well explained. The effectiveness of the proposed framework is validated through both  theoretical analysis and extensive experiments across diverse datasets.

Weakness:
While the paper provides empirical observations that the scattering of anomalous samples tend to be more pronounced in high-dimensional space, it remains unclear whether such a phenomenon generalizes well across datasets with different types of anomalies or varying noise levels. The paper could benefit from systematically examining the scattering phenomenon across more datasets. Additionally, the experimental results should report the mean and standard deviation, with multiple repetitions to ensure statistical reliability.

---

> ### Author Rebuttal · Authors · 2025-07-30
>
> We sincerely thank Reviewer jjez for the detailed comments and insightful questions.
>
> **Q1:** Analysis of the universality of the scattering phenomenon.
>
> To address your concerns about the universality of the scattering phenomenon, we present the quantitative results on the scattering phenomena for all six benchmark datasets used in our study. Furthermore, to evaluate the model's robustness, we inject additive Gaussian noise of three different intensities (σ =0, 0.5, 1.0, 2.0) into the normalized test data (with σ=0.0 as the no-noise baseline). We quantify the impact of noise on the model's internal discriminative power by calculating the average scattering score for normal and anomalous samples, which we define as the mean Euclidean distance of samples to their respective class center, and analyzing their separation ratio.
>
> | Dataset | Noise Level (σ) | Scattering Score (Normal) | Scattering Score (Anomalous) | **Separation Ratio (Anomalous/Normal)** |
> | :------ | :-------------: | :--------------------: | :-----------------------: | :-------------------------------------: |
> | MSL     |       0.0       |          16.0          |           18.2            |                **1.14**                 |
> |         |       0.5       |          21.6          |           31.6            |                **1.46**                 |
> |         |       1.0       |          36.0          |           49.3            |                **1.37**                 |
> |         |       2.0       |          65.9          |           84.6            |                **1.28**                 |
> | PSM     |       0.0       |          31.5          |           37.8            |                **1.20**                 |
> |         |       0.5       |          33.1          |           40.3            |                **1.22**                 |
> |         |       1.0       |          41.2          |           47.4            |                **1.15**                 |
> |         |       2.0       |          61.1          |           65.7            |                **1.08**                 |
> | SWaT    |       0.0       |         73.10          |           94.09           |                **1.29**                 |
> |         |       0.5       |         75.63          |           95.71           |                **1.27**                 |
> |         |       1.0       |         79.22          |           98.03           |                **1.24**                 |
> |         |       2.0       |         84.50          |          102.11           |                **1.21**                 |
> | WADI    |       0.0       |          0.78          |           1.16            |                **1.49**                 |
> |         |       0.5       |          34.9          |           43.4            |                **1.24**                 |
> |         |       1.0       |          87.1          |           72.4            |                  0.83                   |
> |         |       2.0       |         166.7          |           137.0           |                  0.82                   |
> | N-T-S   |       0.0       |          20.2          |           28.4            |                **1.41**                 |
> |         |       0.5       |          21.8          |           32.4            |                **1.49**                 |
> |         |       1.0       |          27.7          |           42.2            |                **1.52**                 |
> |         |       2.0       |          42.1          |           65.5            |                **1.56**                 |
> | N-T-W   |       0.0       |         21.60          |           95.97           |                **4.44**                 |
> |         |       0.5       |         26.10          |          105.33           |                **4.04**                 |
> |         |       1.0       |         40.70          |           41.40           |                **1.02**                 |
> |         |       2.0       |         68.60          |          197.30           |                **2.88**                 |
>
> The results in the table above clearly reveal a consistent anomaly scattering pattern across six different real-world datasets. In addition to this quantitative evidence, we would like to draw the reviewer's attention to Figure 3 in our paper, which qualitatively supports the consistency of the scattering behavior from another perspective. Furthermore, we wish to highlight the case of the WADI dataset, where under medium-to-high noise, the scattering score of normal samples surpasses that of anomalies. We attribute this phenomenon primarily to WADI's high dimensionality (123 dimensions), where strong noise may have a compounding effect and lead to complex changes in inter-variable correlations. This also indicates that while ScatterAD is robust in various noisy environments, its performance can be sensitive to strong noise when dealing with extremely high-dimensional datasets with subtle anomaly patterns. Thank you again for this insightful question, which helps us to more comprehensively characterize the performance boundaries of our method.
>
> **Q2:** Clarification on the reproducibility of experimental results.
>
> Thank you for this question, which is crucial for ensuring the rigor and reproducibility of our work. The experimental results reported in our paper, including the baseline evaluations, are based on the average of three independent runs. We will add standard deviations for all core experimental results in the revised version to more comprehensively demonstrate the stability and statistical reliability of our findings.
>
> **Q3:** Regarding the random initialization strategy for the scattering center $c$.
>
> We sincerely thank you for your question regarding the initialization strategy for the global scattering center $c$. This insightful question is also raised by **Reviewer WWNK**, and we kindly invite you to refer to our detailed response in **Section W1**.  In brief, our experiments show that the random initialization of $c$ is a simple yet effective strategy. For a detailed analysis, including stability tests, we kindly direct you to the **Section W1**.

---

> > ### Comment · Reviewer_jjez · 2025-08-05
> > **Response to Rebuttal**
> >
> > Thank you for the detailed and thoughtful responses. I appreciate the additional quantitative analysis on the universality of the scattering phenomenon, as well as the clarification on experimental reproducibility and initialization strategies.
> >
> > These responses have satisfactorily addressed my concerns, and I believe the authors have made a strong effort to characterize both the strengths and limitations of their approach. Given the current quality and novelty of the work, I am happy to maintain my original evaluation.

---

### Official Review · Reviewer_WWNK · 2025-06-30

**Clarity:** 3
**Significance:** 4
**Originality:** 3
**Rating:** 6
**Confidence:** 5

**Summary:**

The authors insightfully propose ScatterAD, a spatiotemporal anomaly detection framework for industrial IoT, introducing representation dispersion as an inductive bias to model the scattered nature of anomalies in high-dimensional space. ScatterAD combines a topological encoder and a temporal encoder to extract complementary features, and detects anomalies via scattering deviation and temporal inconsistency. The authors also prove that maximizing conditional mutual information is equivalent to a contrastive loss objective. Experiments on six benchmarks confirm its effectiveness through strong performance, visual analysis, and ablation studies.

**Questions:**

Please see weakness.

**Ethical Concerns:**

["NO or VERY MINOR ethics concerns only"]

**Final Justification:**

I carefully read the author's reply and was pleased to find that the author addressed my concerns. Therefore, I am happy to further raise my score. I have no other additional questions.

**Limitations:**

The authors have sufficiently addressed the limitations of their study. I have no additional comments.

**Paper Formatting Concerns:**

No.

**Quality:**

4

**Strengths And Weaknesses:**

Strengths:

The strengths of the work can be summarized as follows:

1. Well written and clear. The paper introduces representation scattering as a novel perspective for high-dimensional anomaly detection, with a logical flow from dispersion to feature separation and detection.
2. The paper presents a novel MTSAD approach inspired by the scattering phenomenon in industrial data, demonstrating clear insight.
3. The effectiveness of the method is demonstrated on multiple MTSAD benchmarks with comprehensive evaluations.
4. The authors’ theory unifies contrastive learning and conditional mutual information to enhance spatiotemporal representation complementarity.
5. The authors provide the code implementation in the supplementary materials, which facilitates reproducibility.

Weaknesses:

1. A potential weakness lies in the initialization of the scattering center 𝑐 in Equation (6), which is randomly sampled within the unit sphere. This strategy may lead to suboptimal local minima and instability during training. For instance, poor initialization is a well-documented problem in machine learning, often causing slower convergence or trapping the model in unfavorable local optima, which degrades performance[1, 2].
2. The scattering mechanism fundamentally depends on cosine similarity between samples and a reference center. However, the manuscript does not discuss the well-documented limitations of cosine similarity in high-dimensional settings, which may affect the robustness of the method.

[1] Glorot, Xavier, and Yoshua Bengio. "Understanding the difficulty of training deep feedforward neural networks." Proceedings of the thirteenth international conference on artificial intelligence and statistics. JMLR Workshop and Conference Proceedings, 2010.

[2] Sutskever, Ilya, et al. "On the importance of initialization and momentum in deep learning." International conference on machine learning. PMLR, 2013.

---

> ### Author Rebuttal · Authors · 2025-07-30
>
> We greatly appreciate Reviewer WWNK for the comprehensive and insightful comments.
>
> **W1**: Regarding the random initialization strategy for the scattering center $c$.
>
> Your concerns about training stability and suboptimal solutions are entirely valid, and these are key considerations in our model design.
>
> First, we wish to clarify the core role of $c$. In our framework, $c$ is initialized only once at the beginning of each training run and **remains fixed throughout the training process**. It is not a parameter learned via backpropagation. Its purpose is twofold:
>
> *   **(1) Provide a fixed anchor point**: $c$ offers a convergent target direction for the representations of normal samples, enabling the model to learn to map these representations into a compact region on the hypersphere.
> *   **(2) Break symmetry and regularize**: If we fix $c$ to a constant (e.g., [1, 0, ..., 0]), it might introduce an undesirable bias, inducing the model to learn a trivial solution where all normal samples map to a specific axis. Randomly initializing $c$ avoids this risk, compelling the model to learn more generalizable representations. This can be viewed as a form of implicit stochastic regularization, preventing the model from overfitting to a pre-defined direction in the latent space.
>
> To empirically address your concerns about instability and suboptimal solutions, we conduct a new experiment: we repeat our full training and evaluation process with 5 different random seeds on the MSL and PSM datasets. A different random seed results in a different scattering center $c$. We report the mean and standard deviation of the Aff-F1 and A-ROC metrics.
>
> | Dataset | Metric | Original | **Mean**  | **Std. Dev.** |
> | :------ | :----- | :------: | :-------: | :-----------: |
> | **MSL** | Aff-F1 |  0.865   | 0.865|    ± 0.003    |
> |         | A-ROC  |  0.986   | 0.986 |    ± 0.001    |
> | **PSM** | Aff-F1 |  0.797   | 0.797 |    ± 0.002    |
> |         | A-ROC  |  0.986   | 0.986 |    ± 0.003    |
>
> Furthermore, we design several variants to compare the performance of **different center initialization strategies**, including:
>
> *   (1) **Zero**: Initialize the center at the origin (zero vector) to test the simplest symmetric starting point.
> *   (2) **Fixed-Radius**: Initialize the center on a hypersphere with a fixed radius to test the model's sensitivity to the initial magnitude.
> *   (3) **Multi-center**: Use multiple independent scattering centers to test the model's ability to capture potentially multi-modal normal data.
>
> In the table below, Num Centers denotes the number of scattering centers used, and Radius specifies the initial magnitude for the fixed_radius strategy. We compare these strategies with the Random-in-Ball (randomly selecting a point within the unit hypersphere) strategy proposed in our paper.
>
> | Dataset | Center Strategy           | Num Centers | Radius |  Aff-F1   |  AUC ROC  |
> | :------ | :------------------------ | :---------: | :----: | :-------: | :-------: |
> | MSL     | **random_in_ball (Ours)** |      1      |  N/A   | 0.867 | **0.986** |
> |         | zero                      |      1      |  0.0   |   0.866   |   0.986   |
> |         | fixed_radius              |      1      |  0.3   |   0.866   |   0.986   |
> |         | fixed_radius              |      1      |  0.7   |   0.864   |   0.986   |
> |         | Multi-center              |      3      |  N/A   |   **0.869**   |   0.986   |
> | PSM     | **random_in_ball (Ours)** |      1      |  N/A   | **0.797** | **0.986** |
> |         | zero                      |      1      |  0.0   |   0.797   |   0.986   |
> |         | fixed_radius              |      1      |  0.3   |   0.797   |   0.986   |
> |         | fixed_radius              |      1      |  0.7   |   0.796   |   0.986   |
> |         | Multi-center              |      3      |  N/A   |   0.792   |   0.980   |
> | SWaT    | **random_in_ball (Ours)** |      1      |  N/A   | **0.704** | **0.982** |
> |         | zero                      |      1      |  0.0   |   0.704   |   0.982   |
> |         | fixed_radius              |      1      |  0.3   |   0.698   |   0.977   |
> |         | fixed_radius              |      1      |  0.7   |   0.702   |   0.979   |
> |         | Multi-center              |      3      |  N/A   |   0.702   |   0.980   |
>
> The results confirm that for most datasets, ScatterAD is not sensitive to the specific initialisation strategy of $c$, and the  **random_in_ball** strategy proposed in our paper is a simple and effective choice.
>
> **W2**: Discussion on the effectiveness of cosine similarity in high-dimensional representation space.
>
> Your point that distances between random vectors in high-dimensional space tend to converge is entirely valid for unprocessed or randomly distributed high-dimensional data. However, the core of our method is to avoid processing such data directly. Instead, the basic task of ScatterAD is to learn a nonlinear mapping from the original high-dimensional space to another well-structured high-dimensional representation space. In this learned space, the distribution of vectors is not random, specifically:
>
> *   (1) $L_{scatter}$: Actively pulls the representations of normal samples towards the same fixed center $c$, forming a compact cluster locally.
> *   (2) $L_{time}$: Enforces temporal continuity in the representation space by minimising the distance between representations of adjacent timesteps.
> *   (3) $L_{contrast}$: Ensures that the representations from different views ($online$  and $target$) remain structurally consistent, further reinforcing the geometry of the representation space.
>
> Therefore, we are not relying on the properties of cosine similarity in a random high-dimensional space; rather, we leverage our model to create a representation space that possesses both spatial structure and temporal continuity. Furthermore, we provide a comparison of our anomaly scoring strategy with two variants based on classic distance metrics:
>
> | Dataset | Metric | ScatterAD (Euclidean - OURS) | ScatterAD (Mahalanobis) | ScatterAD (KL Divergence) |
> | :------ | :----- | :--------------------------: | :---------------------: | :-----------------------: |
> | MSL     | Aff-F1 |          **0.867**           |          0.851          |           0.789           |
> |         | A-ROC  |          **0.986**           |          0.986          |           0.972           |
> | PSM     | Aff-F1 |          **0.797**           |          0.792          |           0.129           |
> |         | A-ROC  |          **0.986**           |          0.980          |           0.793           |
>
> The results show that our simple inductive bias based on Euclidean distance is more effective than methods that attempt to fit complex correlations (Mahalanobis distance) or probability distributions (KL Divergence). This also serves as evidence that our model successfully learns a well-structured representation space.

---

> > ### Comment · Reviewer_WWNK · 2025-08-09
> > **Reply to Author**
> >
> > I carefully read the author's reply and was pleased to find that the author addressed my concerns. Therefore, I am happy to further raise my score. I have no other additional questions.

---

> > > ### Author Response · Authors · 2025-08-09
> > >
> > > Thank you for your positive feedback and for taking the time to review our rebuttal. We are delighted to hear that our response addressed your concerns. Thank you very much for your support and for raising your score.

---

### Official Review · Reviewer_9v8x · 2025-07-02

**Clarity:** 2
**Significance:** 3
**Originality:** 3
**Rating:** 4
**Confidence:** 4

**Summary:**

This paper proposes ScatterAD, a novel approach for multivariate time series anomaly detection that leverages temporal-topological scattering mechanisms. The method combines a temporal encoder and a topological encoder with contrastive fusion to learn complementary spatio-temporal representations. The core insight is that anomalous samples exhibit more pronounced scattering patterns in high-dimensional space compared to normal samples. The authors provide theoretical justification using information bottleneck theory and demonstrate state-of-the-art performance across six real-world datasets.

**Questions:**

(1) Could you provide detailed architectural specifications for the online encoder $f_θ(·)$ and target encoder $f_ϕ(·)$? Additionally, please clarify how h_online and h_target are obtained from these encoders - are they direct outputs or derived through additional processing steps?

(2) Many critical parameters lack proper definition, including $l, k$ in Section 3.2, $τ$ in equation (1), and ξ in equation (8). Could you provide precise definitions of these parameters and their typical value ranges? Furthermore, what are the specific values of parameters such as $l$ you used in your experiments?

(3) How does the theoretical result $I(Z_T; Z_G | G)$ connect to your actual implementation? What specific graph structure $G$ is used in practice, and how does it relate to the conditional mutual information maximization in your theoretical framework?

**Ethical Concerns:**

["NO or VERY MINOR ethics concerns only"]

**Final Justification:**

This paper provides a comprehensive clarification during rebuttal that addresses most of my concerns.

**Limitations:**

Yes

**Paper Formatting Concerns:**

NAN

**Quality:**

2

**Strengths And Weaknesses:**

**Strengths**

(1) The paper introduces conditional mutual information maximization based on information bottleneck theory into multivariate time series anomaly detection, providing theoretical proof that maximizing  $I(Z_T; Z_G \mid G) $ can effectively enhance cross-view consistency and improve representation discriminability.

(2) The method jointly considers temporal features and topological features, significantly enhancing the model's ability to capture complex spatio-temporal dependencies and anomaly patterns.

(3)The paper employs multiple optimization objectives including scattering loss $L_{scatter}$, temporal consistency loss $L_{time}$, and contrastive fusion loss $L_{contrast}$. The contrastive fuse learning design can effectively prevent over-scattering of representations while maintaining temporal structural consistency, simultaneously improving feature discriminability and representation stability.

**Weaknesses**

(1) The paper states that $h_i$  "represent the input feature vectors" in equation (2), but subsequent text indicates that these symbols denote different data representations $h_i = h'_i + h^{(l)}$ while using identical notation. This creates ambiguity regarding what specific data each variable represents.

(2) Section 3.1 introduces the online encoder $f_θ (∙)$ and target encoder $f_ϕ (∙)$ without providing concrete model architecture details. The paper lacks a clear explanation of how $h_{online,t}$ and $h_{target,t}$ are obtained, making it difficult to understand the specific implementation and reproduce the results.

(3) Numerous formulas contain unexplained parameters, including $τ$ in Section 3.2, $l,k$ in equation (1), and ξ in equation (8). Additionally, critical parameter values such as the range or default value of $l,τ$ are not specified.

(4) Section 3.2 states "the node features are aggregated with topological features and temporal features, yielding $h_i = h'_i + h^{(l)}$".
This description lacks clarity regarding how many values are being aggregated and what specific representations correspond to "node features," "topological features," and "temporal features," respectively.

(5) Section 3.3 "Temporal Topological Scattering Representation Learning" presents $c = \text{Randn}(\cdot), c = \frac{c}{\|c\|} \cdot \epsilon$, what appears to be two equations without clearly describing their sequential relationship or dependencies.

(6) The experimental comparison could benefit from including more classic and recent methods such as ModernTCN [Ref1], iTransformer [Ref2], and DCDetector [Ref3].

- [Ref1] Moderntcn: A modern pure convolution structure for general time series analysis, ICLR. 2024.

- [Ref2] Itransformer: Inverted transformers are effective for time series forecasting, ICLR 2023.

- [Ref3]Dcdetector: Dual attention contrastive representation learning for time series anomaly detection, KDD. 2023.

---

> ### Author Rebuttal · Authors · 2025-07-30
>
> We sincerely thank Reviewer 9v8x for the insightful comments and valuable suggestions. We will address your concerns one by one below and will integrate these clarifications into the final version of our paper.
>
> **W1:** Ambiguity of the notation $h_i$.
>
> Thank you for pointing out this ambiguity. We will use $z_i = h_t^t + h_i^{(l)}$ to denote the final representation and will update this notation consistently throughout the rest of the revised manuscript.
>
> **W2:** Lack of specific model architecture details.
>
> To clarify, the online encoder $f_\theta(·)$ and the target encoder $f_\phi(·)$ share the exact same backbone architecture, which consists of a causal convolution module (Eq. 1 in our paper) followed sequentially by a Graph Attention Network (GAT) layer (Eqs. 2 and 3 in our paper). However, the outputs of these two encoders undergo **asymmetric processing steps** before being used in the final loss computation, as detailed below:
>
> | Processing Stage         | Online Path                                   | Target Path                                             |
> | :----------------------- | :-------------------------------------------- | :------------------------------------------------------ |
> | 1. Encoding              | $h_{pred} = f_\theta(X)$                      | $h_{target} = f_\phi(X)$                                |
> | 2. Asymmetric Processing | $h_{online} = \text{Predictor}(h_{pred})$     | $h'\_{target} = L2{Normalize}(h_{target})$ |
> | 3. Loss Contribution     | Used to compute $L_{time}$ and $L_{contrast}$ | Used to compute $L_{scatter}$ and $L_{contrast}$        |
>
> This **asymmetric design** is a well-established and highly effective strategy in momentum contrastive learning frameworks (Kaiming et al., 2020; Byeongchan et al., 2023). It prevents the model from collapsing by breaking the symmetry through an additional learning task (predicting the target representation). We will update the methods section in the final version of our paper with a detailed process description and diagram to ensure the clarity and full reproducibility of our work.
>
> **W3:** Unexplained parameters.
>
> As per your suggestion, we will define all parameters upon their first appearance in the revised manuscript and include their default values. We provide the clarifications here:
>
> *   (1) $\tau$ in Section 3.2: This is the look-back window size of the temporal graph, which defines neighbourhood relations as $(u_{t-k}, u_t)$, where $k \in [1, \tau]$. In our experiments, we use $\tau=2$.
> *   (2) $l$, $k$ in Equation (1): $l$ represents the layer index of the causal convolution, and $k$ is the kernel size. We use two two-layer ($l=2$) with a kernel size of $k=3$.
> *   (3) $\xi$ in Equation (8): This represents the parameters of the target encoder $f_\phi(·)$. We use $\xi$ to maintain consistency with the momentum update literature, where $\theta$ is often used for the online encoder. For clarity, we will unify the notation to use $\phi$.
>
> **W4:** Ambiguity of feature aggregation.
>
> It is important to note that the aggregation operation $h_i = h_t^t + h_i^{(l)}$ is a simple **element-wise addition**. To be precise:
>
> *   (1) $h_t^t$ ("temporal features") is the output of the initial **causal convolution encoder** (from Eq. 1).
> *   (2) $h_i^{(l)}$ ("topological features") are the attention-weighted features learned from neighbouring nodes via the **GAT layer** (from Eq. 3).
> *   (3) $h_i$ ("node features") is the final representation obtained by the **element-wise addition** of these two components.
>
> We will revise the relevant text in Section 3.2 in the revised manuscript, following your suggestion, to explicitly state that this is an element-wise addition and to clarify the origin of each term. Furthermore, as per your suggestion in W1, we will resolve the ambiguity caused by the notation $h_i$.
>
> ---
>
> **W5:** Relationship between the formulas for $c$.
>
> Thank you for requesting clarification on this. The two parts of this formula describe a **two-step sequential process** for initializing the scattering center $c$.
>
> *   (1) Step 1: $c'  = \text{Randn}(\cdot)$: First, a vector $c'$ is sampled from a standard normal distribution.
> *   (2) Step 2: $c = \frac{c'}{\||c'\||} \cdot \epsilon$: The sampled vector $c'$ is then L2-normalized, projecting it onto the unit hypersphere, and subsequently multiplied by a random scalar $\epsilon \in [0, 1)$. This ensures that $c$ is strictly located inside the unit ball.
>
> **W6**: Comparison with classic baselines.
>
> Following your suggestion, we provide extensive comparisons with classic baselines such as ModernTCN, iTransformer, and DCDetector. The results for all baseline models are reproduced by strictly following the configurations and guidelines from their official open-source codebases. Due to space constraints, we present four key scores here; the complete results will be updated in the revised manuscript.
>
> | Dataset       | Metric | ScatterAD | DCDetector | ModernTCN | iTransformer |
> | :------------ | :----- | :-------: | :--------: | :-------: | :----------: |
> | MSL           | Aff-F  | **0.867** |   0.674    |   0.709   |    0.652     |
> |               | PA-F   | **0.964** |   0.957    |   0.807   |    0.659     |
> |               | A-ROC  | **0.986** |   0.961    |   0.627   |    0.604     |
> |               | A-PR   | **0.932** |   0.891    |   0.739   |    0.721     |
> | PSM           | Aff-F  | **0.797** |   0.653    |   0.701   |    0.652     |
> |               | PA-F   | **0.981** |   0.977    |   0.965   |    0.926     |
> |               | A-ROC  | **0.986** |   0.948    |   0.581   |    0.687     |
> |               | A-PR   | **0.969** |   0.866    |   0.629   |    0.751     |
> | SWaT          | Aff-F  |   0.704   |   0.687    |   0.685   |  **0.716**   |
> |               | PA-F   | **0.951** |   0.939    |   0.884   |    0.916     |
> |               | A-ROC  | **0.982** |   0.876    |   0.675   |    0.662     |
> |               | A-PR   | **0.909** |   0.824    |   0.592   |    0.619     |
> | WADI          | Aff-F  |   0.605   | **0.725**  |   0.558   |    0.671     |
> |               | PA-F   | **0.862** |   0.731    |   0.766   |    0.754     |
> |               | A-ROC  | **0.960** |   0.829    |   0.831   |    0.814     |
> |               | A-PR   | **0.765** |   0.651    |   0.697   |    0.627     |
> | NIPS-TS-SWAN  | Aff-F  |   0.038   |   0.484    | **0.533** |    0.507     |
> |               | PA-F   | **0.736** |   0.733    |   0.731   |    0.729     |
> |               | A-ROC  | **0.792** |   0.655    |   0.562   |    0.537     |
> |               | A-PR   | **0.716** |   0.626    |   0.545   |    0.531     |
> | NIPS-TS-GECCO | Aff-F  | **0.825** |   0.648    |   0.446   |    0.435     |
> |               | PA-F   | **0.784** |   0.472    |   0.357   |    0.381     |
> |               | A-ROC  | **0.969** |   0.715    |   0.946   |    0.817     |
> |               | A-PR   | **0.633** |   0.523    |   0.615   |    0.474     |
>
> ---
>
> **Q1:** Encoder architecture specifications and the origin of $h_{online}$, $h_{target}$.
>
> We provide detailed specifications in our response to **W2**. We will also explicitly state these details in Section 3 of the revised manuscript to improve clarity and reproducibility.
>
> **Q2:** Definitions and values of key parameters.
>
> Please see our response to **W3**. We will ensure that all parameters are clearly defined upon their first appearance and will add a table of key hyperparameters in the appendix of the revised manuscript.
>
> **Q3:** Connection between the theoretical result $I(Z_T; Z_G | G)$ and the practical implementation.
>
> Thank you for this crucial question regarding the connection between our mutual information theory and its practical implementation. The link is as follows:
>
> (1) **Theoretical Goal**: Our theory (Appendix A) shows that maximizing the conditional mutual information $I(Z_T; Z_G | G)$ is a principled way to learn complementary representations from the temporal ($Z_T$) and topological ($Z_G$) views, given the graph $G$.
>
> (2) **Practical Implementation**: The **contrastive fusion loss** $L_{contrast}$ (Eq. 7) serves as a tractable, differentiable lower bound for this mutual information. By minimising $L_{contrast}$, we effectively maximise this lower bound, thus driving the model to achieve the theoretical objective. In our implementation, the graph structure $G$ is the **temporal graph described in Section 3.2**, where nodes are time points at timestep $t$, and edges connect temporally adjacent nodes (e.g., $(t-1, t)$). This graph $G$ constrains the entire process, defining the neighbourhood for the GAT's aggregation operation and providing positive pairs for the contrastive loss.
>
> **References**
>
> He, Kaiming, et al. "Momentum contrast for unsupervised visual representation learning." Proceedings of the IEEE/CVF conference on computer vision and pattern recognition. 2020.
>
> Lee, Byeongchan, and Sehyun Lee. "Implicit contrastive representation learning with guided stop-gradient." Advances in Neural Information Processing Systems 36 (2023): 30885-30897.

---

> > ### Comment · Reviewer_9v8x · 2025-08-04
> >
> > I appreciate the clarification from the authors. Now my concerns have been addressed, and I would like to increase the rating.

---

> > > ### Author Response · Authors · 2025-08-05
> > >
> > > We sincerely thank the reviewer for reviewing our rebuttal and for raising your score. Thank you once again for your thoughtful and detailed engagement with our work, which has significantly improved our submission.

---

### Official Review · Reviewer_xfeK · 2025-07-02

**Clarity:** 4
**Significance:** 3
**Originality:** 3
**Rating:** 5
**Confidence:** 3

**Summary:**

This paper presents ScatterAD, a novel anomaly detection framework for multivariate time series in industrial IoT, which links representation learning to the information bottleneck principle, proving that maximizing conditional mutual information between temporal and topological views improves anomaly discrimination. ScatterAD jointly models temporal dynamics and topological techniques via the scattering loss, temporal constraint, and contrastive fusion. Extensive experiments on six benchmarks show state-of-the-art results of ScatterAD.

**Questions:**

1. The temporal graph is fixed during training/inference. I  wonder how the performance of ScatterAD would be if the topology were to change suddenly.
2. Using mean pairwise Euclidean distance instead of dispersion metrics like Mahalanobis distance or others needs clarification.  Does this choice affect sensitivity to feature scaling?
3. PA-F of the method was high, but the AF-F in NIPS-TS-SWAN dropped sharply, although the authors stated the possible reasons in Sec. 4.2. More analysis could reveal method limitations. Or does scattering fail for bursty patterns?

**Ethical Concerns:**

["NO or VERY MINOR ethics concerns only"]

**Final Justification:**

The concerns have been mostly addressed by the authors.

**Limitations:**

1. Although the authors acknowledge static topology and lack of causality modeling in Appendix H. However, these aspects need more expansion.
2. More testing of the proposed method for Real-world noisy data will help explore the robustness.

**Quality:**

3

**Strengths And Weaknesses:**

Strengths: This paper addresses a critical gap in spatio-temporal anomaly detection by unifying temporal consistency and topological dispersion. Achieves impressive results across diverse datasets. It is novel to formalize scattering as a learning signal and integrate information bottleneck theory into multivariate time series anomaly detection. The paper is well-structured, with clear method descriptions and precise theoretical proofs in the Appendix.
Authors provide a technically rigorous study with thorough ablation studies, sensitivity analysis, and visualizations. The proposed scattering mechanism is empirically validated via feature space analysis and training dynamics.

Weaknesses: While inference speed is competitive, the need for careful hyperparameter tuning may hinder plug-and-play adoption in other testing domains. The mutual information bound relies on strong independence assumptions, which may not hold if the graph structure correlates with anomalies. Assuming a static graph structure, it seems to limit its applicability to systems with evolutionary dependencies.

---

> ### Author Rebuttal · Authors · 2025-07-30
>
> We sincerely thank Reviewer xfeK for the detailed and constructive comments, which are instrumental in further improving our paper.
>
> **W:** Regarding hyperparameter sensitivity, theoretical assumptions, and their impact on generalization.
>
> To systematically evaluate the robustness of our model, we conduct a detailed sensitivity analysis in Appendix E of our paper on key hyperparameters, including the time window size, number of encoder layers, hidden dimension, EMA decay rate, and the anomaly threshold $δ$. The results show that ScatterAD maintains stable performance across a wide range of parameter settings. And, we detail the impact of changes in graph structure in Q1.
>
> **Q1:** Regarding the performance of a static graph when the topology changes.
>
> To simulate dynamically changing graph topologies, we extend our data loading process to generate different graph structures for each input time window during training and inference. We design two dynamic topology generation strategies:
>
> (1) **Random Topology**: For each time window, we randomly establish connections between nodes with a certain probability (edge_prob=0.3). To ensure basic graph connectivity, we preserve the connections between adjacent nodes. This strategy simulates scenarios where connections are irregular and appear randomly.
>
> (2) **K-Nearest Neighbours (KNN) Topology**: For the topology constructed within each time window, we compute the Euclidean distance between each node and all other nodes, connecting it to its k-nearest neighbours ($k=3$).
>
> | **MSL**             |   **AFF_F**   |   **PA-F**   |   **A-ROC**   |   **A-PR**   |   **Throughput (samples/second)**   |   **Avg. Inference Time (ms)**   |
> | :-------------- | :-------: | :------: | :-------: | :------: | :--------------------------: | :--------------------------: |
> | Original (Ours)        |   0.866   |  0.964   |   0.986   |  0.932   |            **60.35**             |            **16.571**            |
> | Random Topology |   **0.867**   |  **0.967**   |   **0.987**   |  **0.937**   |            38.33             |            26.091            |
> | KNN Topology    |   0.862   |  0.961   |   0.984   |  0.927   |            49.17             |            20.338            |
> | **PSM**         | **AFF_F** | **PA-F** | **A-ROC** | **A-PR** | **Throughput (samples/second)** | **Avg. Inference Time (ms)** |
> | Original (Ours)         |   0.797   |  **0.981**   |   **0.986**   |  **0.969**   |            **62.17**             |            **16.085**            |
> | Random Topology |   0.791   |  0.975   |   0.980   |  0.961   |            31.77             |            31.472            |
> | KNN Topology    |   **0.803**   |  0.966   |   0.971   |  0.948   |            25.66             |            38.976            |
>
> The results indicate that while dynamic topology strategies lead to marginal performance improvements in some cases (e.g., Random Topology on MSL and KNN Topology on PSM), they introduce additional computational overhead. Therefore, ScatterAD exhibits strong robustness to graph topology, and employing a static graph strikes a good balance between efficiency and effectiveness.
>
> **Q2:** Clarification on the choice of Euclidean distance and its sensitivity to feature scaling.
>
> Our goal is to find a simple yet effective inductive bias. Compared to complex methods that require covariance estimation and rely on distributional assumptions, Euclidean distance, being a computationally simple and assumption-free geometric metric, is a more direct and robust choice to achieve our objective. To address the potential issue of feature scaling,  we first apply standard Z-score normalisation to all input data. Second, we perform L2 normalisation on the target representation $h_{target}$ before calculating the scatter loss ($L_{scatter}$) (Equation 5), ensuring robustness to feature scales.
>
> To address the concerns regarding different distance metrics, we conduct a new experiment comparing our method with two other classic distance metric variants: **Mahalanobis distance** and **Kullback-Leibler (KL) Divergence**. In the Mahalanobis distance variant, we estimate the global covariance matrix from the representations of the training set, following standard practice. We report the performance on three representative datasets below.
>
> | Dataset | Metric | ScatterAD (Euclidean - OURS) | ScatterAD (Mahalanobis) | ScatterAD (KL Divergence) |
> | :------ | :----- | :--------------------------: | :---------------------: | :-----------------------: |
> | MSL     | Aff-F1 |          **0.867**           |          0.851          |           0.789           |
> |         | PA-F   |          **0.964**           |          0.956          |           0.928           |
> |         | A-ROC  |          **0.986**           |          0.986          |           0.972           |
> |         | A-PR   |            0.932             |        **0.934**        |           0.866           |
> | PSM     | Aff-F1 |          **0.797**           |          0.792          |           0.129           |
> |         | PA-F   |          **0.981**           |          0.976          |           0.739           |
> |         | A-ROC  |          **0.986**           |          0.980          |           0.793           |
> |         | A-PR   |          **0.969**           |          0.962          |           0.701           |
>
> **Q3:** In-depth analysis of performance discrepancy on the NIPS-TS-SWAN dataset.
>
> First, we would like to clarify that ScatterAD successfully generates separable anomaly scores for the NIPS-TS-SWAN dataset. The high PA-F1 score (0.736) demonstrates that anomalous events are assigned higher scores than normal ones. Furthermore, to objectively analyse the characteristics of the Aff-F1 and Point-wise F1 metrics, we design a new experiment to simulate the metric sensitivity, which is independent of our model and aims to test the robustness of the Aff-F1 metric itself on the NIPS-TS-SWAN dataset. We assume a model whose predictions can perfectly match the ground truth. We then simulate a minor, realistic localisation error by shifting the entire prediction sequence by $n$ timesteps and observe how the Aff-F1 and Point-wise F1 scores change.
>
> | Dataset      | Localization Error (Shift) | **Aff-F1 Score (Segment-level)** | **Point-wise F1 (Point-level)** |
> | :----------- | :------------------------- | :------------------------------: | :-----------------------------: |
> | NIPS-TS-SWAN | 0 steps                    |            **1.000**             |              1.000              |
> |              | 1 step                     |          0.065 (93.5%↓)          |              0.686              |
> |              | 2 steps                    |              0.174               |              0.697              |
> |              | 5 steps                    |              0.223               |              0.704              |
> |              | 10 steps                   |              0.226               |              0.709              |
> | MSL          | 0 steps                    |            **1.000**             |              1.000              |
> |              | 1 step                     |              1.000               |              0.995              |
> |              | 2 steps                    |              1.000               |              0.991              |
> |              | 5 steps                    |          0.972 (2.8%↓)           |              0.977              |
> |              | 10 steps                   |              0.889               |              0.954              |
>
> The results show that for the NIPS-TS-SWAN dataset, a localisation shift of just one timestep can cause the Aff-F1 score of the model to catastrophically collapse from 1.0 to 0.065. This explains why many other SOTA models (as shown in Table 1 of our paper) also perform poorly on this metric. In contrast, the more traditional Point-wise F1 score shows only a modest decline. This is why we report both scores.
>
> To further validate our model's detection capabilities at a macro level, we compiled overall prediction statistics for ScatterAD on the NIPS-TS-SWAN dataset.
>
> | Statistic                  | Ground Truth | Our Prediction |
> | :------------------------- | :----------: | :------------: |
> | Total Anomaly Ratio        |  **32.6%**   |   **32.1%**    |
> | Number of Anomaly Segments |   551,937    |    843,596     |
>
> The statistics show that the total proportion of anomalies predicted by ScatterAD (32.1%) closely aligns with the ground truth (32.6%), demonstrating its accurate detection capability. However, the higher number of predicted segments suggests that the model may tend to respond to a single continuous anomaly as several smaller segments. Consequently, its performance is underestimated by the Aff-F1 metric under these specific data conditions due to such minor, unavoidable localisation offsets.

---

### Note · Authors · 2025-08-13

We wish to express our sincerest gratitude to the Area Chair and all reviewers for their insightful feedback and constructive engagement throughout the review process.
In our rebuttal, we provide new experiments and analyses to address all raised concerns, including:

(1) Robustness and Generalization: We evaluate ScatterAD's performance with dynamic graph topologies, analyze the universality of the scattering phenomenon under various noise levels, and demonstrate the stability of our initialization strategy across multiple runs and alternative methods.

(2) Methodological Clarity: We clarify our model architecture, parameter definitions, and the connection between our theoretical framework and its practical implementation.

(3) Expanded Evaluation: We add comparisons against several recent state-of-the-art baselines and conduct a deep analysis of metric sensitivity on challenging datasets.

We are pleased that the reviewers find our responses comprehensive and that our efforts to clarify the work are well-received. We are fully committed to integrating all these new results, detailed analyses, and clarifications into the final version of our paper.

---

### Decision · Program_Chairs · 2025-09-17

**Decision:**

Accept (poster)

**Comment:**

This paper makes solid contributions to multivariate time series anomaly detection through: (1) a novel and well-motivated approach using representation scattering, (2) theoretical grounding through information bottleneck theory, (3) strong empirical results with comprehensive evaluation, and (4) clear practical applicability to industrial IoT scenarios. While initial presentation issues were raised, the authors demonstrated commitment to addressing all concerns in the final version. The work represents a meaningful advance in the field with both theoretical insights and practical impact.

The paper received positive evaluations from all four reviewers with final ratings of 5 (Accept), 4 (Borderline Accept), 6 (Strong Accept), and 4 (Borderline Accept). The reviewers appreciated the novel approach of leveraging representation scattering as an inductive signal for multivariate time series anomaly detection, the theoretical grounding through information bottleneck theory, and the comprehensive experimental validation.

While the work is technically solid with good experimental results, it represents an incremental advance in anomaly detection methodology rather than a breakthrough warranting oral/spotlight presentation. The contributions are more suited for detailed technical discussions at a poster session.